# PLATONIC TRANSFORMERS:
# A SOLID CHOICE FOR EQUIVARIANCE

## ABSTRACT

While widespread, Transformers lack inductive biases for geometric symmetries common in science and computer vision. Existing equivariant methods often sacrifice the efficiency and flexibility that make Transformers so effective through complex, computationally intensive designs. We introduce the Platonic Transformer to resolve this trade-off. By defining attention relative to reference frames from the Platonic solid symmetry groups, our method induces a principled weight-sharing scheme. This enables combined equivariance to continuous translations and Platonic symmetries, while preserving the exact architecture and computational cost of a standard Transformer. Furthermore, we show that this attention is formally equivalent to a dynamic group convolution, which reveals that the model learns adaptive geometric filters and enables a highly scalable, linear-time convolutional variant. Across diverse benchmarks in computer vision (CIFAR-10), 3D point clouds (ScanObjectNN), and molecular property prediction (QM9, OMol25), the Platonic Transformer achieves competitive performance by leveraging these geometric constraints at no additional cost.

## 1 INTRODUCTION

Transformers (Vaswani et al., 2017) have become widespread in deep learning, demonstrating unprecedented success on a massive scale (Dosovitskiy et al., 2021; Jumper et al., 2021; Devlin et al., 2019). Their power lies in simple, general-purpose mechanisms that have matured over the years and continue to offer remarkable gains in speed and flexibility, benefiting from vast datasets and computational resources. Yet, this very generality implies they are not inherently equipped to handle specific symmetries present in many scientific domains. For problems with geometric structure, such as those in physics, molecular chemistry, and 3D computer vision, performance can be significantly enhanced by incorporating such inductive bias (Fuchs et al., 2020; Ying et al., 2021; Zhao et al., 2021; Bekkers et al., 2024; Balla et al., 2024; Liao et al., 2024; Romero & Cordonnier, 2021; Wessels et al., 2024; Bose et al., 2024; Zhdanov et al., 2024; Nyholm et al., 2025). The principle of symmetry, for example, has given rise to highly data-efficient and robust group equivariant networks (Cohen & Welling, 2016; 2017; Cesa et al., 2022). However, scaling these symmetry-aware networks has been difficult, as their reliance on operations like group convolutions or Clebsch-Gordan tensor products introduces significant computational overhead compared to standard architectures (He et al., 2021a; Luo et al., 2024). This raises the question: *how can we leverage powerful geometric inductive biases within the transformer architecture without sacrificing the speed and flexibility integral to its success?*

A central challenge in addressing this problem lies in designing an attention mechanism that inherently respects geometric transformations. Such a mechanism would expand on the inductive bias of Transformers, which is typically limited to position embeddings. While widely-used, absolute positional encodings provide location information, but they enforce no explicit relational structure (Shaw et al., 2018; He et al., 2021b). A significant step towards this goal has been the adoption of Rotary Position Embeddings (RoPE) (Su et al., 2024), which endows attention with translation equivariance. Yet, extending this to roto-translation equivariance within the standard Transformer framework remains challenging. Existing approaches often achieve this by making complex architectural changes to equivariant networks that poorly scale or settle for invariant attention mechanisms

which sacrifice feature representations for simplicity and computational efficiency (Masters et al., 2022; Assaad et al., 2023; Thölke & Fabritiis, 2022; Brehmer et al., 2023; Kundu & Kondor, 2025; Joshi et al., 2025). Recent efforts have also explored *hybrid architectures* that resort to symmetry breaking (Qu & Krishnapriyan, 2024; Lawrence et al., 2025) to improve scalability but require a careful mix of modules to maximize downstream performance.

Our main contribution is the *Platonic Transformer*, a framework that achieves equivariance to continuous translations and discrete roto-reflections in Transformers *without changing the underlying attention mechanism or computation graph*. To achieve this, our method processes features relative to a collection of reference frames that form a Platonic symmetry group ($\mathcal{G} \subset O(3)$) and constrains all linear layers to be equivariant with respect to this choice of frame. This principled scheme allows the standard attention block, including its unmodified Rotary Position Embeddings (RoPE), to operate in parallel across these frames, and effectively associates each reference frame with a distinct attention head. As a result, the model incorporates a geometric inductive bias without altering the architecture or computational footprint of a standard Transformer. This enables flexible usage across domains at no additional cost, resolving the long-standing symmetry-awareness vs. scaling dilemma.

Additionally, we analyze the formal connection between RoPE-based attention and convolution to highlight its underlying inductive bias. We show that when the softmax operation is omitted, the attention becomes mathematically equivalent to a dynamic, content-aware convolution. Moreover, in this convolutional setting, the attention operator's complexity scales linearly with the number of tokens, akin to methods like Performer (Choromanski et al., 2020). This result reframes RoPE-attention as a mechanism that explicitly learns and applies dynamic, content-aware geometric filters.

## 2 BACKGROUND: TRANSFORMERS WITH POSITION EMBEDDINGS

The core of a Transformer is its self-attention mechanism, which computes outputs for a sequence of input features $\{f_i \in \mathbb{R}^C\}$ based on pairwise interactions. To perform spatial tasks, this operation must incorporate the position $p_i \in \mathbb{R}^n$ associated with each feature $f_i$. This positional information, often added via absolute or relative encodings, allows the model to learn relationships that respect geometric symmetries.

### 2.1 VANILLA ATTENTION AND ABSOLUTE POSITIONING

Given a sequence of input features $\mathbf{f}_i \in \mathbb{R}^C$, the self-attention layer first computes query, key, and value vectors via linear projections: $\mathbf{q}_i = \mathbf{W}^Q \mathbf{f}_i$, $\mathbf{k}_j = \mathbf{W}^K \mathbf{f}_j$, $\mathbf{v}_j = \mathbf{W}^V \mathbf{f}_j$. Here, the learnable weight matrices are $\mathbf{W}^Q, \mathbf{W}^K \in \mathbb{R}^{C \times d}$ and $\mathbf{W}^V \in \mathbb{R}^{C \times C'}$. The output for the $i$-th feature, $\mathbf{y}_i \in \mathbb{R}^{C'}$, is a weighted sum of the value vectors, with weights determined by softmax-normalized dot products of queries and keys:

$$\mathbf{y}_i = \sum_{j=1}^{N} \mathrm{attn}(\mathbf{q}_i, \mathbf{k}_j)\mathbf{v}_j \,, \quad \text{where} \quad \mathrm{attn}(\mathbf{q}_i, \mathbf{k}_j) = \mathop{\mathrm{softmax}}_{j} \left(\mathbf{q}_i^\top \mathbf{k}_j\right) . \tag{1}$$

As this operation is permutation-equivariant, it is insensitive to the order of the inputs and must be modified to incorporate positional information for spatial tasks. A common approach is to use Absolute Positional Encodings (APE), where a unique vector $\mathbf{E}(\mathbf{p}_i)$ is added to each input feature, $\mathbf{f}_i' = \mathbf{f}_i + \mathbf{E}(\mathbf{p}_i)$, before the linear projections are applied. The attention score is then computed from these position-aware features. However, since this interaction depends on absolute coordinates rather than relative positions, APE is not translation-equivariant.

### 2.2 ROTARY POSITION EMBEDDINGS (RoPE)

RoPE achieves a more structured approach to position encoding (Su et al., 2024). Instead of adding a positional vector, RoPE modifies the query and key vectors with a position-dependent transformation, making the attention score explicitly dependent on relative positions.

This transformation is constructed by stacking 2D rotation matrices, giving RoPE its name. To apply RoPE with positions $\mathbf{p}$ in dimension $n > 1$, we use a set of $n$-dimensional frequency vectors

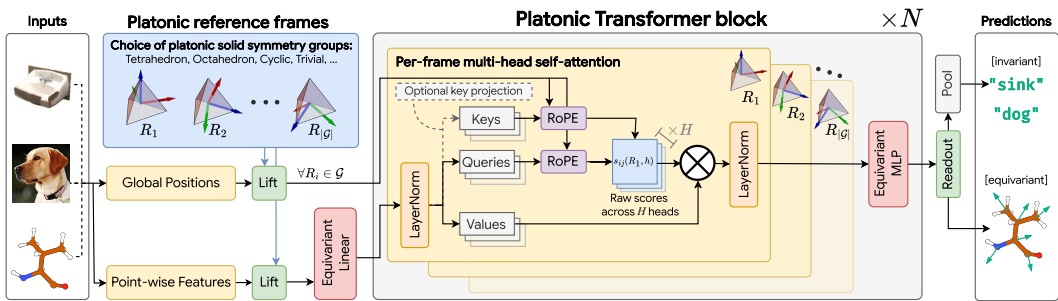

Figure 1: Visualization of Weight-Shared RoPE within the $N$-layer Platonic Transformer. Scalar and vector inputs are lifted to become functions on the platonic solid symmetry group of choice (here, the Tetrahedral group). The same multi-head self-attention mechanism is applied in parallel, with each instance rotating the features according to a different reference frame $R_i \in \mathcal{G}$. Choosing the trivial group as $\mathcal{G}$ reduces this framework to a standard Transformer.

$\Omega = \{\boldsymbol{\omega}_k\}_{k=1}^{d/2}$, each defining a direction used to project $\mathbf{p}$ to 1D and a frequency used to apply 1D-RoPE in this direction. We obtain $d/2$ blocks,

$$\rho_{\boldsymbol{\omega}_k}(\mathbf{p}) = \begin{pmatrix} \cos(\boldsymbol{\omega}_k^\top \mathbf{p}) & -\sin(\boldsymbol{\omega}_k^\top \mathbf{p}) \\ \sin(\boldsymbol{\omega}_k^\top \mathbf{p}) & \cos(\boldsymbol{\omega}_k^\top \mathbf{p}) \end{pmatrix}, \tag{2}$$

which are stacked in a block-diagonal manner to form a single transformation matrix, $\boldsymbol{\rho}_\Omega(\mathbf{p})$:

$$\boldsymbol{\rho}_\Omega(\mathbf{p}) = \mathrm{diag}(\rho_{\boldsymbol{\omega}_1}(\mathbf{p}), \dots, \rho_{\boldsymbol{\omega}_{d/2}}(\mathbf{p})). \tag{3}$$

Note that while $\boldsymbol{\rho}_\Omega(\mathbf{p})$ is a high-dimensional rotation, this rotation is not related to rotations of the position $\mathbf{p}$. In fact, $\boldsymbol{\rho}_\Omega$ is instead connected to translations of $\mathbf{p}$, formally discussed in Appendix A.

For a query $\mathbf{q}_i$ at position $\mathbf{p}_i$ and a key $\mathbf{k}_j$ at position $\mathbf{p}_j$, $\boldsymbol{\rho}_\Omega$ is applied before the dot product. As the operator $\boldsymbol{\rho}_\Omega$ is orthogonal and satisfies the homomorphism property[1] for translations, the interaction simplifies to depend only on relative positions:

$$\left(\boldsymbol{\rho}_\Omega(\mathbf{p}_i)\mathbf{q}_i\right)^\top \left(\boldsymbol{\rho}_\Omega(\mathbf{p}_j)\mathbf{k}_j\right) = \mathbf{q}_i^\top \boldsymbol{\rho}_\Omega(\mathbf{p}_i)^\top \boldsymbol{\rho}_\Omega(\mathbf{p}_j)\mathbf{k}_j = \mathbf{q}_i^\top \boldsymbol{\rho}_\Omega(\mathbf{p}_j - \mathbf{p}_i)\mathbf{k}_j. \tag{4}$$

This final form reveals the core property of RoPE. Although widely adopted for its empirical success, the mechanism's effectiveness is not coincidental; it directly embeds translation equivariance into the attention mechanism by making the score a function of content and relative positions. This powerful geometric inductive bias, often hidden within the standard Transformer framework, provides a principled reason for RoPE's strong performance (Chen et al., 2023; Dai et al., 2019). The formal construction of this operator from the first principles of group theory is detailed in Appendix A.

## 3 THE PLATONIC TRANSFORMER

We generalize the principle of RoPE to obtain equivariance not only under continuous translations, but also discrete roto-reflections. We obtain roto-reflection equivariance by redefining the positional encoding relative to a set of reference frames defined as elements in a discrete subgroup $\mathcal{G} \subset O(n)$. Traditional RoPE-attention operates on a single global reference frame. Instead, we perform attention on multiple frames in parallel. A key advantage of our method is that it leaves the rope-attention mechanism and the overall computation graph unchanged from the traditional transformer.

### 3.1 FEATURES RELATIVE TO REFERENCE FRAMES

Throughout the architecture, features are represented and processed *relative to the reference frames* defined by the elements of a discrete group $\mathcal{G} \subset O(n)$. Since input features are typically defined in a

---

[1]The operator $\boldsymbol{\rho}_\Omega(\mathbf{p})$ being orthogonal means its inverse is its transpose: $\boldsymbol{\rho}_\Omega(\mathbf{p})^{-1} = \boldsymbol{\rho}_\Omega(\mathbf{p})^\top$. The homomorphism property for the translation group satisfies $\boldsymbol{\rho}_\Omega(\mathbf{p_i} + \mathbf{p_j}) = \boldsymbol{\rho}_\Omega(\mathbf{p_i})\boldsymbol{\rho}_\Omega(\mathbf{p_j})$.

global frame of reference, they must first be *lifted* to become functions on the group $\mathcal{G}$. Specifically, each feature becomes a map $\mathbf{f}_i(\cdot) : \mathcal{G} \to \mathbb{R}^C$, where $\mathbf{f}_i(R)$ is the feature vector at point $i$ viewed from frame $R \in \mathcal{G}$. For the finite groups we consider, this map is represented as a tensor of shape $[|\mathcal{G}|, C]$. We denote this tensor simply as a flattened vector $\mathbf{f}_i \in \mathbb{R}^{|G| \cdot C}$ and use the functional notation $\mathbf{f}_i(\cdot)$ to emphasize its role as a feature map. As we will see, the flattened vector viewpoint is key to preserving the standard Transformer computation graph.

The lifting process depends on the geometric type of the input feature. Scalar features, being invariant to viewpoint, are lifted to constant functions by copying them across all frames. Vector features, in contrast, are expressed relative to each frame; for example, a single 3D vector feature $\mathbf{u} \in \mathbb{R}^3$ is lifted to a three-channel signal on the group via the transformation $\mathbf{f}(R) = R^{-1}\mathbf{u}$. All such lifted components can be concatenated, after which they are processed by the subsequent equivariant, frame-dependent attention layers.

## 3.2 Weight-sharing across RoPE Embeddings

The key step for achieving equivariance to $\mathcal{G}$ as well as translations is making the RoPE operator itself dependent on the reference frames. This is achieved by projecting the position $\mathbf{p}_i$ of each input token $i$ onto $R$, which yields views $\mathbf{p}_i(R) = R^{-1}\mathbf{p}_i$ relative to each frame. As the queries $\mathbf{q}_i$, keys $\mathbf{k}_j$, and values $\mathbf{v}_j$ are obtained by applying equivariant linear projections (cf. Section 3.3) to the feature maps $\mathbf{f}_i$, they are also functions on the group. We can then compute the unnormalized attention scores from the perspective of frame $R$, which we denote as $s_{ij}(R)$:

$$s_{ij}(R) = \mathbf{q}_i(R)^\top \rho_\Omega(\mathbf{p}_j(R) - \mathbf{p}_i(R))\mathbf{k}_j(R) \tag{5}$$

$$= \mathbf{q}_i(R)^\top \rho_\Omega((\mathbf{p}_j - \mathbf{p}_i)(R))\mathbf{k}_j(R) \,. \tag{6}$$

Scores for each frame are computed *in parallel* as their own independent attention head. Note that we can also obtain $s_{ij}(R)$ by steering the base set of frequencies $\Omega$ instead of the positions $\mathbf{p}_i$, which we show in Appendix D. However, from our current perspective, the RoPE-attention mechanism itself remains completely unchanged from its traditional formulation in Eq. 4; only the relative positions $\mathbf{p}_i - \mathbf{p}_j$ are now defined relative to each reference frame $R$. The attention coefficients are obtained by applying the softmax to the scores $s_{ij}(R)$. The output $\mathbf{y}_i(R)$ for each token $i$ is then given as

$$\mathbf{y}_i(R) = \sum_{j=1}^N \mathrm{attn}_{ij}(R)\mathbf{v}_j(R), \quad \text{where} \quad \mathrm{attn}_{ij}(R) = \operatorname*{softmax}_j(s_{ij}(R)) \,. \tag{7}$$

This process naturally results in an output tensor $\mathbf{y}_i \in \mathbb{R}^{|\mathcal{G}| \cdot C}$, where the features are defined relative to each frame. Notably, the base frequencies $\Omega$ of RoPE are shared across frames and this leads to the operator being equivariant to the roto-reflections in $G$, as we detailed in Appendix B.

## 3.3 Equivariant Linear Layers and Fixed Computation Graph

All linear transformations, including the query, key, and value projections ($\mathbf{W}^Q, \mathbf{W}^K, \mathbf{W}^V$), and any MLP blocks, must be equivariant. As our features can be viewed either as functions on the group, $\mathbf{f}_i(\cdot)$, or as flattened vectors, $\mathbf{f}_i \in \mathbb{R}^{|\mathcal{G}| \cdot C}$, we can describe the action of an equivariant linear layer $\Phi$ from both perspectives. From the flattened vector viewpoint, the layer is a standard matrix-vector multiplication, $\mathbf{y}_i = \mathbf{W}\mathbf{f}_i$. However, for this transformation to be equivariant, the weight matrix $\mathbf{W}$ cannot be arbitrary; it must have a specific, constrained structure.

The equivariance constraint is defined from the functional viewpoint: for any group element $R \in \mathcal{G}$, the transformation must satisfy $\Phi(L_R\mathbf{f}_i) = L_R(\Phi(\mathbf{f}_i))$, where $L_R$ is the action of rotating the reference frames, i.e., $(L_R\mathbf{f}_i)(\tilde{R}) = \mathbf{f}_i(R^{-1}\tilde{R})$. This constraint is satisfied *if and only if* the layer's operation is a *group convolution* (Cohen et al., 2019, Thm. 3.1). This gives the layer a dual identity: it is a convolution over the group axis, which is mathematically equivalent to a matrix-vector multiplication with a structured, weight-shared matrix:

$$(\Phi(\mathbf{f}_i))(R) := \sum_{\tilde{R} \in \mathcal{G}} \mathbf{W}_{\text{group}}(R^{-1}\tilde{R})\,\mathbf{f}_i(\tilde{R}) \quad \Longleftrightarrow \quad \Phi(\mathbf{f}_i) := \mathbf{W}\mathbf{f}_i \,. \tag{8}$$

Here, $\mathbf{W}_{\text{group}} : \mathcal{G} \to \mathbb{R}^{C' \times C}$ is a learnable kernel defined on the group. The large matrix $\mathbf{W} \in \mathbb{R}^{(|\mathcal{G}| \cdot C') \times (|\mathcal{G}| \cdot C)}$ is a block matrix whose blocks are determined by the kernel values: $[\mathbf{W}]_{R, \tilde{R}} =$

$\mathbf{W}_{\text{group}}(R^{-1}\tilde{R})$. This structure imposes a weight-sharing scheme where the interaction between input and output frames depends only on their *relative pose*, $R^{-1}\tilde{R}$. The layer is thus constrained to learn patterns from the geometric arrangement of features, rather than their absolute pose.

While the group convolution formulation makes the geometric inductive bias explicit, the matrix-vector viewpoint clarifies that this is in essence a principled *weight-sharing scheme* that preserves the computation graph of a standard linear layer (we're still doing matrix-vector multiplication). A favorable side-effect, however, is that this structure reduces the parameter count from the $(|\mathcal{G}| \cdot C') \times (|\mathcal{G}| \cdot C)$ of an unconstrained layer to just $|\mathcal{G}| \cdot C' \cdot C$—a reduction by a factor of $|\mathcal{G}|$.

Crucially, by choosing the number of channels $C$ such that the effective feature dimension $C \cdot |\mathcal{G}|$ is held constant, the overall matrix dimensions are identical regardless of the group size. The trivial group $\mathcal{G} = \{e\}$ illustrates the base case, where the operation collapses to a standard linear layer with a weight matrix $\mathbf{W}_{\text{group}}(e)$ of size $C' \times C$. The geometric inductive bias is therefore not introduced by adding new, complex modules, but by imposing a structure on the weights of existing ones.[2]

With all components of the architecture now defined as equivariant operations, we can formally state the key property of the full model, namely equivariance under the discrete group $\mathcal{G} \subset O(n)$.

**Proposition 1** (End-to-End Equivariance). *Our proposed Transformer architecture is an equivariant model. A global roto-reflection $R \in \mathcal{G}$ applied to the input point cloud results in a corresponding transformation $L_R$ of the final output feature maps.*

The proof is given in Appendix B.

### 3.4 Frame Selection Via Platonic Solids

The final step is to select a suitable subgroup $\mathcal{G} \subset O(n)$ to serve as the reference frames. We select them from the discrete symmetry groups of regular polygons and polyhedra, with different considerations for 2D and 3D as illustrated in Figure 2.

*In 3D*, we restrict our frames to the finite *rotational* symmetry groups ($\mathcal{G} \subset SO(3)$) of the Platonic solids: the *tetrahedral* (12 rotations), *octahedral* (24 rotations), and *icosahedral* (60 rotations) groups. While these solids have larger full symmetry groups that include reflections (e.g., 24 total symmetries for the tetrahedron), we focus on the purely rotational subgroups for a more tractable structure.

*In 2D*, we consider discrete subgroups of $O(2)$, which correspond to the symmetries of regular polygons. This includes both the rotation-only *cyclic groups* ($C_n$) and the *dihedral groups* ($D_n$), which contain both rotations and reflections. Here $n$ denotes the group's order. This discrete subgroup approach is advantageous for two reasons. First, it provides a finite set of frames that forms a structured and approximately uniform discretization of the underlying continuous spaces of orientations ($SO(3)$ in 3D and $O(2)$ in 2D). Second, and more critically, these frames form a group. This is essential for maintaining a meaningful geometric structure, as it ensures that layers can operate equivariantly, keeping features coherently defined relative to our chosen frames throughout the network.

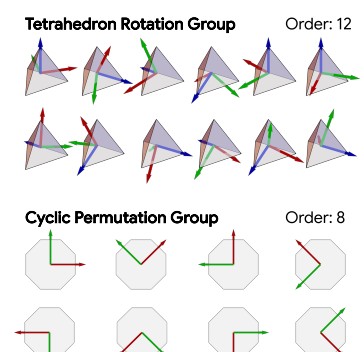

Figure 2: Elements of the symmetry groups of platonic solids form a subgroup of $SO(3)$.

The advantage of working with a *finite* group $\mathcal{G}$ is that its operations can be handled discretely and efficiently using *Cayley tables*. We assign a unique index $i \in \{0, \ldots, |\mathcal{G}| - 1\}$ to each rotation $R_i \in \mathcal{G}$. The group product $R_i R_j = R_k$ can then be precomputed and stored in the Cayley table, a simple look-up table where $\text{Cayley}[i, j] = k$. This discrete formalism makes the group action on our feature maps, which are functions on the group $f : \mathcal{G} \to \mathbb{R}^C$, extremely efficient. A rotation of this feature map by an element $R_i$, defined by the action $(L_{R_i} f)(R_j) = f(R_i^{-1} R_j)$, simplifies to a permutation of the feature tensor's entries. With

---

[2]This structure can even give computational benefits, by implementing the linear layers in the Fourier domain of $\mathcal{G}$ (Bökman et al., 2025). In Appendix N, we find that at marginally higher channel counts than used in this paper, a Fourier implementation leads to greatly improved training throughput, indicating that this is a promising direction for future research.

the Cayley table, the new feature at position $j$ is simply copied from the old feature at position $k = \text{Cayley}[\text{inverse}[i], j]$.

# 4 INDUCTIVE BIAS OF PLATONIC TRANSFORMERS

This section examines the Platonic Transformer's structural inductive biases. We highlight its interpretation as a dynamic group convolution and its equivariant attention, contrasting these with approaches based on invariant attention.

## 4.1 PLATONIC TRANSFORMER AS DYNAMIC GROUP CONVOLUTION

The use of RoPE in a linear attention setting establishes a deep connection to convolution. Specifically, the mechanism implements an adaptive convolution where the kernel is synthesized on-the-fly. This dynamic kernel is expressed as an expansion in a sparse Fourier basis, defined by the RoPE frequencies, and the coefficients for this basis expansion are provided by the query vectors. This makes the convolution content-aware. We formalize this as follows (proof in Appendix C.1).

**Proposition 2** (Linear RoPE Attention as Dynamic Convolution). *Consider a standard linear attention layer using RoPE with constant key vectors ($\mathbf{k}_j = \mathbf{1}$). The layer's output $\mathbf{y}_i$ is mathematically equivalent to a dynamic convolution:*

$$\mathbf{y}_i = \sum_{j=1}^{N} \phi_{\mathbf{q}_i}(\mathbf{p}_j - \mathbf{p}_i)\mathbf{v}_j \,, \tag{9}$$

*where the dynamic kernel $\phi_{\mathbf{q}_i}$ is given by the inverse sparse Fourier transform:*

$$\phi_{\mathbf{q}_i}(\Delta\mathbf{p}) = \sum_{k=1}^{d/2} \left[ a_k(\mathbf{q}_i)\cos(\boldsymbol{\omega}_k^{\top}\Delta\mathbf{p}) + b_k(\mathbf{q}_i)\sin(\boldsymbol{\omega}_k^{\top}\Delta\mathbf{p}) \right] \,. \tag{10}$$

*The Fourier coefficients are given by the linear projections $a_k(\mathbf{q}_i) = q_{i,2k-1} + q_{i,2k}$ and $b_k(\mathbf{q}_i) = q_{i,2k} - q_{i,2k-1}$, where $q_{i,m}$ is the $m$-th element of the query vector $\mathbf{q}_i$.*

**Remark 1** (Purely Geometric vs. Mixed Kernels). *This result recasts the query's role: rather than simply probing for content, $\mathbf{q}_i$ enables the parameters to construct a unique geometric filter. The formulation of the key vector is a design choice. The constant-key formulation ($\mathbf{k}_j = \mathbf{1}$) forces the model to learn purely geometric, content-adaptive convolution operators. In contrast, a learned key ($\mathbf{k}_j = \mathbf{W}^K \mathbf{f}_j$) results in a mixed kernel whose coefficients depend on both query and key features, and thus entangles geometry and signal.*

Our choice of the purely geometric formulation was motivated by a key experimental observation: training with a mixed kernel from learned keys was highly unstable on the QM9 and OMol25 datasets (cf. Appendix L). We hypothesize this instability arises because these tasks require learning universal *physical principles* that are purely functions of geometry, whereas computer vision tasks like ScanObjectNN involve learning *statistical correlations* between local appearance and global shape. A mixed kernel *entangles* these universal principles with instance-specific features, creating an unstable optimization problem with *conflicting gradients* as the model attempts to learn a general physical law while simultaneously fitting unique local chemical environments. In computer vision, this same entanglement is beneficial, as learning the statistical interplay between features and geometry is the primary objective. Fixing the keys thus acts as a crucial regularizer for physical systems by forcing the model to prioritize the disentangled geometric principles, and thus stabilizing the training process. The convolution perspective further leads to a key practical advantage.

**Corollary 1** (Linear-Time Complexity). *The dynamic convolution in Proposition 2 can be computed in $O(N)$ time, where $N$ is the number of tokens or points in the point cloud. This offers a scalable alternative to standard attention, which has a quadratic complexity of $O(N^2)$.*

Within our Platonic Transformer, this entire mechanism is lifted to operate over the reference frames defined by a group $\mathcal{G}$. Consequently, the operator becomes an adaptive *group convolution* (proof in Appendix C.3), where the kernel is steered by the group elements/reference frames.

### 4.2 Invariant vs. Equivariant Attention Score

Our approach implements an *equivariant attention* mechanism, where the attention pattern is orientation-dependent. This contrasts with methods using an *invariant attention* score, which applies the same pattern from all orientations (Fuchs et al., 2020; Chen & Villar, 2022; Assaad et al., 2023; Frank et al., 2024; Knigge et al., 2024; Kundu & Kondor, 2025; Nordström et al., 2025).

For multi-head attention with $H$ heads, let $\mathbf{q}_i(R, h)$, $\mathbf{k}_j(R, h)$, and $\mathbf{v}_j(R, h)$ denote the projected query, key, and value vectors for head $h$ from the perspective of frame $R$. In our equivariant approach, the raw scores $s_{ij}(R, h)$ are passed directly to the softmax. This allows the model to learn orientation-dependent attention patterns, making it a more expressive formulation that retains the rich geometric information in the features. The output is an equivariant feature map on the group:

$$\mathbf{y}_i(R, h) = \sum_{j=1}^{N} \operatorname*{softmax}_j(\underbrace{s_{ij}(R, h)}_{R-\text{dependent}})\mathbf{v}_j(R, h), \;\; s_{ij}(R, h) = \mathbf{q}_i(R, h)^{\top}\boldsymbol{\rho}_{\Omega_h}((\mathbf{p}_j - \mathbf{p}_i)(R))\mathbf{k}_j(R, h).$$

(11)

In practice, this is efficiently implemented by treating the $|\mathcal{G}|$ perspectives as an independent set of attention heads. Tensors are reshaped so that the group and head dimensions are merged, e.g., to a shape of $[B, N, |\mathcal{G}| \cdot H, C_h]$, before the dot product calculation.

In an invariant attention score, a single attention pattern is created by pooling the raw scores over the group axis before the softmax, akin to the *symmetrization* in the RoPE-based approach of Frank et al. (2024). These invariant attention scores are then applied to the original equivariant value vectors. The resulting output is still equivariant, but it is derived from an *orientation-agnostic attention pattern*:

$$\mathbf{y}_i(R, h) = \sum_{j=1}^{N} \operatorname*{softmax}_j(\underbrace{s_{ij}^{\text{inv}}(h)}_{R-\text{agnostic}})\mathbf{v}_j(R, h), \quad \text{where} \quad s_{ij}^{\text{inv}}(h) = \sum_{R \in \mathcal{G}} s_{ij}(R, h).$$

(12)

Although simpler, this formulation sacrifices the model's ability to attend to features in an orientation-dependent manner. Implementing Eq. 12 can be done by reshaping tensors so that the group and channel dimensions are merged, to shape $[B, N, H, |\mathcal{G}| \cdot C_h]$, as then the dot-product in $s_{ij}$ and the sum in $s_{ij}^{\text{inv}}$ are simultaneously computed when taking the dot-product between queries and keys. For a fully invariant output, one could additionally average the value vectors $\mathbf{v}_j(R, h)$ over the group to further collapse the geometric representation.

## 5 Experiments

To validate our proposed architecture, we conduct a series of experiments across a number of different tasks and datasets. Our evaluation is structured to analyze the role of the equivariance inductive bias by categorizing tasks into two distinct settings based on their inherent geometric properties.

First, for tasks with inherent symmetry, such as those in QM9 (Ramakrishnan et al., 2014) and OMol25 (Levine et al., 2025), the underlying molecular systems have no canonical orientation. Their properties are determined by the relative positions of atoms and are independent of the global coordinate system. Since the physical laws governing these molecular properties are E(3)-symmetric, equivariance becomes a fundamental requirement for a model to generalize efficiently (Fuchs et al., 2020; Bronstein et al., 2021; Batzner et al., 2022; Pacini et al., 2025; Vadgama et al., 2025). We refer to this category of problems as *Equivariant Tasks*.

Second, for tasks involving datasets with a canonical orientation, like CIFAR-10 (Krizhevsky, 2009) and ScanObjectNN (Uy et al., 2019), strict end-to-end equivariance is not required (the images/objects are aligned w.r.t. a canonical up-direction). These problems nevertheless provide a testbed to investigate if the geometric inductive bias of our model, enforced by weight-sharing, improves performance on its own merits. We refer to these as *Non-Equivariant Tasks*.

### 5.1 Experimental Setup

All Platonic Transformer variants are built upon RoPE, making them inherently *translation-equivariant*. The degree of rotational equivariance is then determined by the choice of a discrete

symmetry group $\mathcal{G} \subset O(n)$ that defines the set of reference frames. For instance, selecting the trivial group ($\mathcal{G} = \emptyset$) results in a purely translation-equivariant model ($T(n)$); it uses only the identity frame. Choosing the rotational symmetry group of the Tetrahedron provides 12 reference frames, making the model approximately $SE(n)$-equivariant, or $E(n)$-equivariant when including reflections too.

For fair comparison, we match the computational cost between $SE(n)$ and $T(n)$ models by equating our group-based parallelism with standard multi-head attention. For instance, an $SE(n)$ model using the 12-element tetrahedral group with one head per frame is benchmarked against a $T(n)$ baseline with 12 total heads (details in App. G). For certain tasks, symmetries can be conditionally broken by using APE or providing an *external reference frame*, yet internal layers critically retain principled weight-sharing. Using external frames to break symmetry and APE to provide geometric information are effective strategies, allowing a model to benefit from geometric inputs without being end-to-end constrained by full equivariance (Vadgama et al., 2025).

For CIFAR-10 and ScanObjectNN, we conducted a comprehensive sweep to find the optimal configuration. In contrast, for QM9 and OMol25, we used a sequential process: first, we identified the best architecture via an extensive sweep on QM9, then transferred these hyperparameters to OMol25 for further refinement with a one-million subset before full training (see Appendix H-K).

## 5.2 Non-Equivariant Tasks

**Cifar10**  The results of our ablation study on CIFAR-10 are presented in Table 1. The findings indicate that incorporating 2D rotational symmetries provides a tangible benefit over the translation-only baseline (the $\emptyset*$ model, which is equivalent to a standard Vision Transformer). This suggests that even for general-purpose vision tasks without an end-to-end equivariance requirement, equivariance proves to be an important inductive bias. This may be explained by the fact that even though images have a canonical pose (e.g. with the sky at the top), equivariance allows for internal weight-sharing and thus the reuse of patterns (edges, parts, objects) that may appear at arbitrary orientations within an image. The comparison between the full attention and linear-convolutional shows a significant impact of attention over the linear complexity dynamic convolution counterpart (in which the softmax is omitted, cf. Prop. 2).

**ScanObjectNN**  On the ScanObjectNN point cloud classification task, we test the effectiveness of 3D symmetry groups (trivial vs tetrahedron vs horizontal flips) in a realistic setting with occlusions and significant orientation variability. The results, shown in Table 2 again highlight the impact of equivariance and weightsharing. While the quadratic-cost attention mechanism offers greater expressive power, the linear-time convolutional variant provides a significant speed-up, which is critical for efficiently processing large point clouds. This demonstrates the versatility of our approach in adapting to different computational and modeling requirements in 3D computer vision, as demonstrated in Figure 3. Also note that the computational cost is independent of the chosen symmetry group.

Table 1: CIFAR-10 Accuracy (%).

| Group | Attention Acc. (↑) | Conv Acc. (↑) | # Params |
|---|---|---|---|
| $\emptyset*$ | 91.49 | 88.56 | 85.1M |
| $C_4$ | **92.73** | **88.70** | 21.3M |
| $C_6$ | 92.16 | 88.44 | 14.2M |
| $D_4$ | 92.53 | 88.10 | 21.3M |
| $D_6$ | 92.07 | 88.58 | 14.2M |
| Flop | 91.49 | 87.86 | 42.6M |

Table 2: ScanObjectNN Overall Acc. (%).

| Group | Attention Acc. (↑) | Conv Acc. (↑) |
|---|---|---|
| $\emptyset*$ | 80.5 | 79.8 |
| Tetrahedron | 81.3 | 80.1 |
| Flop | **82.0** | **80.6** |

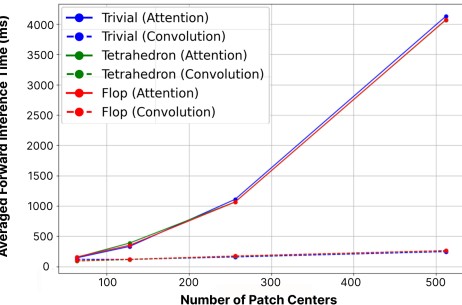

Figure 3: The Platonic Transformer, when configured in its convolutional mode, exhibits a linear computational complexity relative to the input sequence length, a complexity shared with its attention mode. It is noteworthy that the model's specific equivariance type does not alter this computational scaling.

Table 3: QM9 Property Prediction MAE ($\downarrow$).

**Platonic Transformer (end-to-end)**

| Group | Attention | | Convolution | |
|---|---|---|---|---|
| | $\mu$ | $\alpha$ | $\mu$ | $\alpha$ |
| $\emptyset$ | 0.028 | 0.064 | 0.030 | 0.061 |
| Tetrahedron | 0.012 | 0.049 | 0.014 | 0.047 |
| Octahedron | **0.010** | **0.048** | 0.012 | 0.047 |

**Platonic Transformer with PCA-based frames**

| Method | | | | |
|---|---|---|---|---|
| 8-Refl + 1 frame | 0.039 | 0.155 | – | – |

**Reference methods**

| Method | *(re)produced in this work | | | |
|---|---|---|---|---|
| EquiformerV2 [38] | **0.010** | 0.050 | – | – |
| FAFormer [28]* | 0.122 | 0.252 | – | – |
| G-Hyena [42]* | – | – | 0.018 | 0.066 |
| Rapidash [62] | – | – | **0.010** | **0.040** |

Table 4: Inference wall-clock times on QM9.

**Platonic Transformer**

| Group | Avg. Time (ms) ($\downarrow$) |
|---|---|
| $\emptyset$ | $2.87 \pm 0.29$ |
| Tetrahedron | $\mathbf{2.79} \pm 0.21$ |
| Octahedron | $2.85 \pm 0.25$ |

**Reference methods**

| Method | |
|---|---|
| Standard Transformer | $\mathbf{2.01} \pm 3.74$ |
| G-Hyena [42] | $44.06 \pm 60.05$ |
| TFN [58] | $590.45 \pm 269.25$ |

Table 5: OMol25 Energy/Force MAE ($\downarrow$).

Our own (re)produced results, (4 GPUs - 120 hrs)

| Method | Force | Energy | E/Atom |
|---|---|---|---|
| Platonic Transformer | 24.25 | **74.00** | **2.63** |
| eSEN [36] | **23.92** | 120.0 | 3.37 |

From literature, (estimated 4 GPUs - 475 hrs)

| eSEN [36] | **10.11** | **29.80** | **0.88** |
|---|---|---|---|
| MACE [36] | 16.83 | 54.09 | 1.55 |

## 5.3 EQUIVARIANT TASKS

**QM9** Our results on the QM9 benchmark are summarized in Table 7. The first group of results identifies the most effective Platonic group and model variant for this task, showing a performance gain from incorporating Platonic symmetries over the translation-only ($\emptyset$ group) baseline. Both the Tetrahedron and Octahedron groups achieve strong performance, delivering results on par with state-of-the-art methods like EquiformerV2 (Liao et al., 2024). Our linear-convolutional models are similarly effective, outperforming other convolution-based baselines such as G-Hyena (Moskalev et al., 2025), a state-space model for long-context geometric modeling. Our end-to-end $SE(3)$-equivariant model is superior to existing baselines, demonstrating that a directly learned geometric representation is fundamentally more effective compared to external symmetrization.

Our use of reference frames relates to the popular class of "frame-based methods," which are a popular approach for building equivariant networks Du et al. (2023); Yin et al. (2025); Puny et al. (2022). One such prominent framework is Frame Averaging (FA), which incorporates group equivariance via *symmetrization* (detailed introduction in Appendix E) of non-equivariant neural network backbones. FA can be extended to full $E(3)$ by including roto-reflections (e.g., roto-reflecting PCA axes) but comes at a cost, requiring a separate forward pass for each frame element. Our Platonic Transformer offers a more scalable drop-in replacement. For instance, using just a single orthogonal matrix from PCA (i.e., just one frame element) along with an axis-aligned roto-reflection group (here, we use $C_2 \times C_2 \times C_2$, a subgroup of the octahedral group) achieves native $E(3)$ equivariance at an $\sim 8x$ lower cost. We evaluate the effectiveness of FA by comparing to FAFormer (Huang et al., 2024), which applies FA to a standard Transformer backbone with 8 frame elements. Our single-frame variant with only octahedral reflections (`8-Refl + 1 frame`) outperforms FAFormer with a smaller compute cost. This underscores the need for a geometrically expressive backbone, such as the Platonic Transformer.

**OMol25** To validate the scalability and performance of our proposed architecture, we evaluate our model with the best hyperparameters on the large-scale OMol25 dataset. For a fair comparison under a constrained computational budget, we compare the Platonic Transformer with eSEN (Levine et al., 2025), the current state-of-the-art method on OMol25, for 120 hours on a node with 4 NVIDIA 6000Ada GPUs; note that we use the hyperparameters for eSEN indicated in the original work. The results are presented in Table 5, demonstrating that our model significantly outperforms the baseline under these identical conditions for energy prediction and achieves highly competitive performance in force prediction. This performance is noteworthy when contextualized against current literature benchmarks, which we estimate from Levine et al. (2025), utilized a computational budget nearly four times larger. Achieving strong results under such constraints indicates the architectural efficiency of our model, suggesting that the Platonic Transformer could likely achieve state-of-the-art accuracy with a comparable computational budget which we leave for future work.

**Inference times**  Given that the Platonic Transformer does not alter the computation graph of the standard Transformer, our method benefits from similarly fast inference speeds. A single Platonic Transformer layer runs at the same order of magnitude as a standard Transformer layer (implemented using a single `TransformerEncoderLayer` module in PyTorch) on a batch size of 64 molecules from QM9 on a single H200 GPU, averaged over 10 batches, as shown in Table 4. We also show superior inference times, by 2-3 orders of magnitude, against a single G-Hyena and Tensor Field Network (Thomas et al., 2018) layer under the same setup while still retaining $E(3)$ equivariance.

## 6 CONCLUSION

We introduce the Platonic Transformer, a framework that achieves approximate $E(n)$ equivariance without compromising the flexibility and scalability of the standard Transformer architecture. By combining Rotary Position Embeddings (RoPE) with a new frame-dependent attention mechanism— where attention is computed relative to reference frames from Platonic solid symmetry groups—we integrate a powerful geometric inductive bias while preserving the original computation graph and cost. This approach demonstrates that principled equivariance and modern scalability are not mutually exclusive. Furthermore, our analysis reveals a formal equivalence to dynamic group convolution with linear complexity, enabling a highly scalable, linear-time variant for large-scale tasks. In many scientific domains, equivariance represents a "Platonic ideal" — an essential physical principle a model should respect. By eliminating the trade-off between this principled design and computational efficiency, the Platonic Transformer makes this ideal a practical and scalable reality.

## REPRODUCIBILITY STATEMENT

We make several strides towards reproducibility of our work. We back our theoretical results with proofs in Appendix C, and with rough intuitions in the main text. We have attached a `.zip` folder as supplementary material with source code to train our Platonic Transformer models (and baselines) across different tasks and datasets; we also provide adequate comments and documentation. Details required to reproduce the results in our tables and figures are provided in Appendices H-K. We also provide information on hyperparameter-tuning efforts in Appendices F and G. We intend to open-source the code on public platforms like GitHub once the review period has formally ended.

## ETHICS STATEMENT

Our work addresses fundamental challenges in building scalable methods for deep learning and AI for Science. We believe there is potential for our methods to be used in scientific domains like biotechnology as well as energy and sustainability. As our work is in its early stages, we posit that it introduces dedicated computational methods rather than focusing on particular applications that may warrant closer oversight and caution.

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

# Appendix

## Table of Contents

# A  ROTARY POSITION EMBEDDINGS FROM A GROUP THEORETICAL PERSPECTIVE

A fundamental challenge in geometric deep learning is creating position representations that respect underlying symmetries. For data in $\mathbb{R}^d$, our goal is to define a high-dimensional position embedding, $\mathbf{E} : \mathbb{R}^d \rightarrow \mathbb{R}^{d'}$, that is equivariant to translations. This requires that for any translation vector $\mathbf{p}$, the embedding transforms predictably: $\mathbf{E}(\mathbf{p}_0 + \mathbf{p}) = \boldsymbol{\rho}(\mathbf{p})\mathbf{E}(\mathbf{p}_0)$, where $\boldsymbol{\rho}(\mathbf{p})$ is a linear transformation. Group representation theory provides the formal tools to construct such embeddings.

## A.1  THE THEORETICAL TOOLKIT

To proceed, we first define the essential concepts required for our construction.

**Definition 1** (Representation). *A linear representation of a group $\mathcal{G}$ on a vector space $V$ is a group homomorphism $\rho : \mathcal{G} \rightarrow GL(V)$, where $GL(V)$ is the general linear group of invertible linear transformations on $V$.*

To ensure that the positional encoding does not arbitrarily amplify or diminish feature magnitudes, which would destabilize learning, we require the representations to be length-preserving. This leads to the concept of a unitary representation.

**Definition 2** (Unitary Representation). *A representation $\rho$ is **unitary** if it maps group elements to unitary operators, i.e., $\rho : \mathcal{G} \rightarrow U(V)$. For real-valued representations, this corresponds to orthogonality, $\rho(g)^{-1} = \rho(g)^\top$.*

Just as a complex signal can be decomposed into pure frequencies, a general representation can be broken down into fundamental building blocks known as irreducible representations (irreps).

**Definition 3** (Irreducible Representation). *An **irreducible representation (irrep)** is a representation acting on a vector space $V$ that has no non-trivial invariant subspaces.*

## A.2  CONSTRUCTING THE ROPE OPERATOR

With these formal tools, we can now build the RoPE operator. The irreps of the translation group $(\mathbb{R}^d, +)$ are indexed by a frequency vector $\boldsymbol{\omega} \in \mathbb{R}^d$ and are given by one-dimensional, unitary representations:

$$\rho_{\boldsymbol{\omega}}(\mathbf{p}) = e^{i\boldsymbol{\omega}^\top \mathbf{p}}. \tag{13}$$

This exponential form is the unique continuous solution to the group's homomorphism property, $\rho(\mathbf{p}_1 + \mathbf{p}_2) = \rho(\mathbf{p}_1)\rho(\mathbf{p}_2)$, where the imaginary exponent ensures unitarity.

However, neural networks typically operate on real numbers. We can obtain a real-valued irrep by combining pairs of conjugate frequencies, $\boldsymbol{\omega}_k$ and $-\boldsymbol{\omega}_k$. This yields a 2D irreducible representation that takes the familiar form of a rotation matrix:

$$\rho_{\boldsymbol{\omega}_k}(\mathbf{p}) = \begin{pmatrix} \cos(\boldsymbol{\omega}_k^\top \mathbf{p}) & -\sin(\boldsymbol{\omega}_k^\top \mathbf{p}) \\ \sin(\boldsymbol{\omega}_k^\top \mathbf{p}) & \cos(\boldsymbol{\omega}_k^\top \mathbf{p}) \end{pmatrix}. \tag{14}$$

To create a high-dimensional embedding, we simply select a set of frequencies $\Omega = \{\boldsymbol{\omega}_k\}_{k=1}^{d'/2}$ and stack these 2D rotation blocks along the diagonal of a larger matrix. This results in a single, block-diagonal transformation that correctly and equivariantly updates the entire embedding for a given translation $\mathbf{p}$:

$$\boldsymbol{\rho}_\Omega(\mathbf{p}) = \mathrm{diag}(\rho^{\boldsymbol{\omega}_1}(\mathbf{p}), \ldots, \rho_{\boldsymbol{\omega}_{d'/2}}(\mathbf{p})). \tag{15}$$

This is the core mechanism behind Rotary Position Embeddings. Its structure guarantees both equivariance and computational efficiency, as each 2D component can be rotated independently.

**Definition 4** (Rotary Position Embedding (RoPE) Operator). *The RoPE operator $\boldsymbol{\rho}_\Omega(\mathbf{p})$ for a position $\mathbf{p} \in \mathbb{R}^d$ is the block-diagonal rotation matrix defined above (Equation 15), constructed from a set of frequencies $\Omega$. The application of RoPE to a feature vector $\mathbf{f} \in \mathbb{R}^{d'}$ is defined as the matrix-vector product: $\boldsymbol{\rho}_\Omega(\mathbf{p})\mathbf{f}$. For this operation to be well-defined, the feature dimension $d'$ must be even.*

## A.3 TRANSLATION INVARIANCE IN ATTENTION

While the RoPE operator provides an *equivariant* transformation for feature vectors, its crucial benefit within the Transformer architecture is that it makes the attention score *invariant* to global translations. This property ensures that the attention mechanism only considers the relative positions of tokens, which is the inductive bias we seek. We formalize this key result below.

**Proposition 3** (Translation Invariance of the RoPE Attention Score). *The attention score computed using RoPE,* $\mathrm{attn}(\mathbf{q}, \mathbf{k}, \Delta\mathbf{p}) = \mathbf{q}^\top \boldsymbol{\rho}_\Omega(\Delta\mathbf{p})\mathbf{k}$, *where* $\Delta\mathbf{p} = \mathbf{p}_j - \mathbf{p}_i$, *is invariant to a global translation of the coordinate system.*

*Proof.* Let the positions $\mathbf{p}_i$ and $\mathbf{p}_j$ be translated by an arbitrary vector $\mathbf{t} \in \mathbb{R}^d$, resulting in new positions $\mathbf{p}'_i = \mathbf{p}_i + \mathbf{t}$ and $\mathbf{p}'_j = \mathbf{p}_j + \mathbf{t}$. The new relative displacement vector, $\Delta\mathbf{p}'$, is:

$$\Delta\mathbf{p}' = \mathbf{p}'_j - \mathbf{p}'_i = (\mathbf{p}_j + \mathbf{t}) - (\mathbf{p}_i + \mathbf{t}) = \mathbf{p}_j - \mathbf{p}_i = \Delta\mathbf{p}. \tag{16}$$

Since the relative displacement vector is unchanged by the global translation, the RoPE operator applied to it also remains unchanged: $\boldsymbol{\rho}_\Omega(\Delta\mathbf{p}') = \boldsymbol{\rho}_\Omega(\Delta\mathbf{p})$. Consequently, the attention score, which depends only on the content vectors and this operator, is invariant to the translation:

$$\mathbf{q}^\top \boldsymbol{\rho}_\Omega(\Delta\mathbf{p}')\mathbf{k} = \mathbf{q}^\top \boldsymbol{\rho}_\Omega(\Delta\mathbf{p})\mathbf{k}. \tag{17}$$

This formally demonstrates that RoPE imparts translation invariance to the attention mechanism. $\square$

## A.4 A FOURIER PERSPECTIVE

The principle of constructing equivariant functions from irreducible representations is deeply connected to Fourier analysis. The Fourier transform provides a way to decompose any function on a group into a weighted sum (or integral) over its irreps. For the translation group on $\mathbb{R}^d$, these irreps are precisely the complex exponentials we used as our building blocks. Therefore, RoPE can be understood as a practical application of Fourier theory, using a discrete basis of Fourier modes (the chosen frequencies $\Omega$) to represent the positional signal.

**Definition 5** (Fourier Transform on $\mathbb{R}^d$). *The forward Fourier transform* $\mathcal{F} : L^2(\mathbb{R}^d) \to L^2(\mathbb{R}^d)$ *maps a function* $f$ *to its frequency-space representation* $\hat{f}$. *The coefficient for a frequency* $\boldsymbol{\omega}$ *is the projection of* $f$ *onto the corresponding irrep* $\rho_{\boldsymbol{\omega}}$:

$$\hat{f}(\boldsymbol{\omega}) = \mathcal{F}\{f\}(\boldsymbol{\omega}) = \int_{\mathbb{R}^d} f(\mathbf{p})\overline{\rho_{\boldsymbol{\omega}}(\mathbf{p})} \, d\mathbf{p} = \int_{\mathbb{R}^d} f(\mathbf{p})e^{-i\boldsymbol{\omega}^\top \mathbf{p}} \, d\mathbf{p}. \tag{18}$$

*The inverse transform reconstructs the function by integrating over all irreps:*

$$f(\mathbf{p}) = \mathcal{F}^{-1}\{\hat{f}\}(\mathbf{p}) = \frac{1}{(2\pi)^d} \int_{\mathbb{R}^d} \hat{f}(\boldsymbol{\omega})\rho_{\boldsymbol{\omega}}(\mathbf{p}) \, d\boldsymbol{\omega} = \frac{1}{(2\pi)^d} \int_{\mathbb{R}^d} \hat{f}(\boldsymbol{\omega})e^{i\boldsymbol{\omega}^\top \mathbf{p}} \, d\boldsymbol{\omega}. \tag{19}$$

# B EQUIVARIANCE PROPERTIES OF PLATONIC TRANSFORMERS

We formally establish the equivariance of our proposed architecture. We consider a point cloud $\{\mathbf{p}_i, \mathbf{v}_i, s_i\}_{i=1}^N$ consisting of positions, vectors, and scalars. A global rotation $R \in \mathcal{G}$ acts on these inputs as $\mathbf{p}_i \mapsto R\mathbf{p}_i$, $\mathbf{v}_i \mapsto R\mathbf{v}_i$, and $s_i \mapsto s_i$.

**Equivariant Feature Lifting.** Input features are first lifted to functions on the group $\mathcal{G}$. The lifting operator, $\mathrm{Lift}$, maps the input point cloud to a set of feature maps $\{\mathbf{f}_i : \mathcal{G} \to \mathbb{R}^C\}_{i=1}^N$. Scalar components are copied to each frame, while vector components (from $\mathbf{p}_i, \mathbf{v}_i$) are lifted by projecting them onto each reference frame. This projection means expressing the vector's coordinates in the local basis of a given frame $\tilde{R} \in \mathcal{G}$, which is achieved by the transformation $\tilde{R}^{-1}\mathbf{v}$. This lifting procedure is equivariant by construction: a global rotation $R$ of the input point cloud results in the lifted feature maps transforming via the left regular representation, $L_R$. That is:

$$(\mathrm{Lift}(R \cdot \mathrm{cloud}))_i(\tilde{R}) = (\mathrm{Lift}(\mathrm{cloud}))_i(R^{-1}\tilde{R}) \triangleq (L_R \mathbf{f}_i)(\tilde{R}). \tag{20}$$

**Equivariant Linear Layers.** All linear layers $\Phi$ in our network are implemented as point-wise group convolutions, as shown in Eq. 8. These layers are equivariant to the action of the group by construction (Cohen et al., 2019, Thm. 3.1), satisfying $\Phi(L_R\mathbf{f}_i) = L_R(\Phi(\mathbf{f}_i))$.

This leads to our main proposition regarding the attention mechanism.

**Proposition 4** (Equivariant Attention). *Let the queries $Q_i$, keys $K_i$, and values $V_i$ be equivariant feature maps produced by the equivariant linear layers. The RoPE-enhanced attention mechanism (Eq. 7), which computes outputs $\mathbf{y}_i$, is an equivariant operation. That is, if the inputs transform as $\mathbf{f}_i \mapsto L_R\mathbf{f}_i$, the outputs transform as $\mathbf{y}_i \mapsto L_R\mathbf{y}_i$.*

*Proof.* Let the inputs to the attention layer $(Q_i, K_i, V_i)$ transform under a global rotation $R$ as $Q_i' = L_RQ_i$, $K_i' = L_RK_i$, and $V_i' = L_RV_i$. We analyze the transformation of each component of the attention calculation.

The score function $s_{ij}(\tilde{R})$, which depends on $Q_i(\tilde{R})$ and $K_j(\tilde{R})$ (and potentially RoPE terms derived from lifted positions), will transform as:

$$s_{ij}'(\tilde{R}) = \text{score}(Q_i'(\tilde{R}), K_i'(\tilde{R}), \dots) = \text{score}(Q_i(R^{-1}\tilde{R}), K_j(R^{-1}\tilde{R}), \dots) = s_{ij}(R^{-1}\tilde{R}).$$

This means the score function itself is equivariant, $s_{ij}' = L_R s_{ij}$. Since the softmax operator is applied point-wise for each frame $\tilde{R}$ over the index $j$, the attention weights also transform equivariantly:

$$\text{attn}_{ij}'(\tilde{R}) = \text{softmax}_j(s_{ij}'(\tilde{R})) = \text{softmax}_j(s_{ij}(R^{-1}\tilde{R})) = \text{attn}_{ij}(R^{-1}\tilde{R}).$$

Finally, the output feature map $\mathbf{y}_i$ transforms as:

$$\mathbf{y}_i'(\tilde{R}) = \sum_{j=1}^{N} \text{attn}_{ij}'(\tilde{R})V_j'(\tilde{R})$$

$$= \sum_{j=1}^{N} \text{attn}_{ij}(R^{-1}\tilde{R})V_j(R^{-1}\tilde{R}) = \mathbf{y}_i(R^{-1}\tilde{R}).$$

Thus, the output transforms as $\mathbf{y}_i' = L_R\mathbf{y}_i$, proving the attention mechanism is equivariant. $\square$

## C  PROOFS

### C.1  PROOF OF PROPOSITION 2

We seek to show that the unnormalized attention score, which defines the kernel $\phi_{\mathbf{q}_i}(\Delta\mathbf{p})$, takes the form of a sparse Fourier series whose coefficients are linear projections of the query $\mathbf{q}_i$.

Let the relative position be $\Delta\mathbf{p} = \mathbf{p}_j - \mathbf{p}_i$. With a constant key vector $\mathbf{k}_j = \mathbf{1}$, the kernel is defined by the attention score:

$$\phi_{\mathbf{q}_i}(\Delta\mathbf{p}) = (\boldsymbol{\rho}(\mathbf{p}_i)\mathbf{q}_i)^\top (\boldsymbol{\rho}(\mathbf{p}_j)\mathbf{1})$$

Using the properties of the RoPE operator $\boldsymbol{\rho}$, this simplifies to:

$$\phi_{\mathbf{q}_i}(\Delta\mathbf{p}) = \mathbf{q}_i^\top \boldsymbol{\rho}(\mathbf{p}_i)^\top \boldsymbol{\rho}(\mathbf{p}_j)\mathbf{1} = \mathbf{q}_i^\top \boldsymbol{\rho}(\Delta\mathbf{p})\mathbf{1}$$

The RoPE matrix $\boldsymbol{\rho}(\Delta\mathbf{p})$ is block-diagonal, consisting of $d/2$ independent 2D rotation blocks. We can therefore analyze the contribution from a single block $k$ and sum the results. Let $\theta_k = \boldsymbol{\omega}_k^\top \Delta\mathbf{p}$. The contribution from block $k$ is:

$$\phi_k = \begin{pmatrix} q_{2k-1} & q_{2k} \end{pmatrix} \begin{pmatrix} \cos(\theta_k) & -\sin(\theta_k) \\ \sin(\theta_k) & \cos(\theta_k) \end{pmatrix} \begin{pmatrix} 1 \\ 1 \end{pmatrix}$$

Performing the matrix-vector multiplications, we get:

$$\phi_k = \begin{pmatrix} q_{2k-1} & q_{2k} \end{pmatrix} \begin{pmatrix} \cos(\theta_k) - \sin(\theta_k) \\ \sin(\theta_k) + \cos(\theta_k) \end{pmatrix}$$

$$= q_{2k-1}(\cos(\theta_k) - \sin(\theta_k)) + q_{2k}(\sin(\theta_k) + \cos(\theta_k))$$

Grouping terms by $\cos(\theta_k)$ and $\sin(\theta_k)$ reveals the linear projections:

$$\phi_k = \underbrace{(q_{2k-1} + q_{2k})}_{a_k(\mathbf{q}_i)} \cos(\theta_k) + \underbrace{(q_{2k} - q_{2k-1})}_{b_k(\mathbf{q}_i)} \sin(\theta_k)$$

The Fourier coefficients $a_k(\mathbf{q}_i)$ and $b_k(\mathbf{q}_i)$ are thus simple linear combinations of the query vector's elements. Summing over all $k = 1, \ldots, d/2$ yields the complete kernel $\phi_{\mathbf{q}_i}(\Delta \mathbf{p})$, which has the exact form stated in the proposition.

## C.2 PROOF OF COROLLARY 1

The linear-time complexity is achieved by expressing the operation in matrix form and re-ordering the computation. Let $\mathbf{q}_i' = \boldsymbol{\rho}_\Omega(\mathbf{p}_i)\mathbf{q}_i$ and $\mathbf{k}_j' = \boldsymbol{\rho}_\Omega(\mathbf{p}_j)\mathbf{1}$. Let $\mathbf{Q}' \in \mathbb{R}^{N \times d'}$ be the matrix with rows $(\mathbf{q}_i')^\top$, $\mathbf{K}' \in \mathbb{R}^{N \times d'}$ be the matrix with rows $(\mathbf{k}_j')^\top$, and $\mathbf{V} \in \mathbb{R}^{N \times d_v}$ be the matrix of value vectors. The output matrix $\mathbf{Y} \in \mathbb{R}^{N \times d_v}$ is given by:

$$\mathbf{Y} = (\mathbf{Q}'(\mathbf{K}')^\top)\mathbf{V}.$$

By the associativity of matrix multiplication, this can be computed as $\mathbf{Y} = \mathbf{Q}'((\mathbf{K}')^\top \mathbf{V})$. The term $(\mathbf{K}')^\top \mathbf{V}$ costs $O(Nd'd_v)$ to compute, resulting in a $d' \times d_v$ matrix. Multiplying this by $\mathbf{Q}'$ costs an additional $O(Nd'd_v)$. The total complexity is therefore $O(Nd'd_v)$, linear in the sequence length $N$.

## C.3 PROOF OF PLATONIC TRANSFORMERS IMPLEMENTING GROUP CONVOLUTIONS

The dynamic convolution from Proposition 2 becomes a dynamic *group convolution* within the Platonic Transformer. This is a direct consequence of applying the operation to lifted coordinates $\mathbf{p}_i(R) = R^{-1}\mathbf{p}_i$ for each reference frame $R \in \mathcal{G}$. Since the relative position vector becomes $R^{-1}(\mathbf{p}_j - \mathbf{p}_i)$, the kernel's input is transformed accordingly. The resulting output for each frame takes the form of a group cross-correlation[3]:

$$\mathbf{y}_i(R) = \sum_{j=1}^{N} \phi_{\mathbf{q}_i(R)} \left( R^{-1}(\mathbf{p}_j - \mathbf{p}_i) \right) \mathbf{v}_j(R). \tag{21}$$

Here, the kernel $\phi_{\mathbf{q}_i(R)}$ is steered by the group element $R$, defining an equivariant dynamic group convolution.

# D EQUIVALENT ATTENTION VIA RoPE BASE FREQUENCY STEERING

We achieve full equivariance to Euclidean transformations by making the RoPE operator dependent on a local reference frame $R$ by projecting positions $\mathbf{p}_i$ on $R$ to obtain positions $\mathbf{p}_i(R) := R^{-1}\mathbf{p}_i$. The attention scores $s_{ij}(R)$ for a query $\mathbf{q}(R)_i$ and key $\mathbf{k}(R)_j$ are computed as:

$$s_{ij}(R) = \mathbf{q}_i(R)^\top \boldsymbol{\rho}_\Omega((\mathbf{p}_j - \mathbf{p}_i)(R))\mathbf{k}_j(R), \tag{22}$$

An equivalent approach is to steer the set of base RoPE frequencies $\Omega$ for each frame, creating a frame-specific set $\Omega_R = \{R\boldsymbol{\omega}_k \mid \boldsymbol{\omega}_k \in \Omega\}$ (Reddy & Chatterji, 1996). The attention scores are then computed as:

$$\hat{s}_{ij}(R) = \mathbf{q}_i(R)^\top \boldsymbol{\rho}_{\Omega_R}(\mathbf{p}_j - \mathbf{p}_i)\mathbf{k}_j(R). \tag{23}$$

*Proof.* For $s_{ij}(R)$ and $\hat{s}_{ij}(R)$ to be equivalent, we require that $\boldsymbol{\rho}_\Omega((\mathbf{p}_j - \mathbf{p}_i)(R)) = \boldsymbol{\rho}_{\Omega_R}(\mathbf{p}_j - \mathbf{p}_i)$. For this, we need to show that $\boldsymbol{\omega}_k^\top \Delta \mathbf{p}(R) = (R\boldsymbol{\omega})_k^\top \Delta \mathbf{p}$, where $\Delta \mathbf{p} = \mathbf{p}_j - \mathbf{p}_i$. Let $\mathbf{R}$ be the orthogonal matrix corresponding to $R$. Then we have:

$$\boldsymbol{\omega}_k^\top \Delta \mathbf{p}(R) = \boldsymbol{\omega}_k^\top R^{-1} \Delta \mathbf{p} \tag{24}$$

$$= \boldsymbol{\omega}_k^\top \mathbf{R}^\top \Delta \mathbf{p} \tag{25}$$

$$= (\mathbf{R}\boldsymbol{\omega}_k)^\top \Delta \mathbf{p} \tag{26}$$

$$= (R\boldsymbol{\omega}_k)^\top \Delta \mathbf{p} \tag{27}$$

Thus, projecting global positions or steering the base frequencies are equivalent. $\square$

---

[3]Following common convention, we refer to this operation as a group convolution, though it is technically a cross-correlation Cohen & Welling (2016); Bekkers (2020).

By projecting the global positions, the RoPE attention mechanism remains identical to its traditional formulation. Steering the base frequencies, however, is often more computationally efficient, since the number of base frequencies is typically much smaller than the input sequence length.

## E  FRAME AVERAGING

Frame Averaging (Puny et al., 2022) (FA) imbues symmetry awareness to arbitrary neural networks. A backbone model $\Phi : V \to W$ on normed spaces can be made equivariant (or invariant) to a group $\mathcal{G}$. Specifically, for two group representations $\rho_1(g)$ and $\rho_2(g)$ that $g \in \mathcal{G}$ induces on $V$ and $W$, $\Phi(\rho_1(g) \cdot \mathbf{x}) = \rho_2(g) \cdot \Phi(\mathbf{x})$, where $\mathbf{x} \in V$ and $\cdot$ is the group action on elements from $V$ and $W$.

FA first constructs a *frame* $\mathcal{F}(\mathbf{x}) : V \to 2^{\mathcal{G}}$ with the following properties:

- A frame is $\mathcal{G}$-equivariant if $\mathcal{F}(\rho_1(g)\mathbf{x}) = g\mathcal{F}(\mathbf{x})$, where $g\mathcal{F}(\mathbf{x}) = \{gh | h \in \mathcal{F}(\mathbf{x})\}$, and
- A frame is bounded over a domain $K \subset V$ if $\exists c > 0$ such that the operator norm $\|\rho_2(g)\|_{\mathrm{op}} \leq c, \forall g \in \mathcal{F}(\mathbf{x}), \forall \mathbf{x} \in K$.

To make the backbone $\Phi$ $\mathcal{G}$-equivariant, *symmetrization* is applied on $\mathbf{x}$ via the group averaging operator:

$$\langle \Phi \rangle_{\mathcal{F}}(\mathbf{x}) = \frac{1}{|\mathcal{F}(\mathbf{x})|} \sum_{g \in \mathcal{F}(\mathbf{x})} \rho_2(g)\Phi(\rho_1(g)^{-1}(\mathbf{x})). \tag{28}$$

For equivariance to 3-dimensional Euclidean rigid-body transformations (translations, rotations, reflections), i.e., $\mathcal{G} := E(3)$ and $V = \mathbb{R}^3$, $\mathcal{F}(\mathbf{x})$ is constructed using PCA on $\mathbf{x}$; this involves computing the centroid $\mathbf{t} = \frac{1}{n}\mathbf{x}^{\top}\mathbf{1} \in \mathbb{R}^3$ and covariance matrix $\mathbf{C} = (\mathbf{x} - \mathbf{1}\mathbf{t}^{\top})^{\top}(\mathbf{x} - \mathbf{1}\mathbf{t}^{\top})$ for a point cloud $\mathbf{x} \in \mathbb{R}^{n \times 3}$ with $n$ points. Suppose we obtain eigenvectors $\mathbf{u}_1, \mathbf{u}_2, \mathbf{u}_3 \in \mathbb{R}^3$ of $\mathbf{C}$, we construct $3 \times 3$ orthogonal matrices by concatenating them together. Depending on the collection of these matrices, we achieve equivariance to different motion groups: for $E(3)$ equivariance, we use $\mathbf{U} = [\pm\mathbf{u}_1, \pm\mathbf{u}_2, \pm\mathbf{u}_3] \subset E(3)$, and if we restrict this collection to contain only orthogonal, positive orientation matrices, we achieve $SE(3)$ equivariance. The frame then looks like,

$$\mathcal{F}(\mathbf{x}) = \{(\mathbf{U}, \mathbf{t}) : \mathbf{U} = [\pm\mathbf{u}_1, \pm\mathbf{u}_2, \pm\mathbf{u}_3])\} \subset E(3).$$

For equivariant predictions on atomistic point clouds, we set $\rho_1(g)\mathbf{x} = \mathbf{x}\mathbf{U}^{\top} + \mathbf{1}\mathbf{t}^{\top}$ and $\rho_2(g)\mathbf{x} = \mathbf{x}\mathbf{U}^{\top}$ for each frame element, followed by the application of the averaging operator in Eq. 28. Furthermore, Puny et al. (2022) show that if $\mathcal{F}(\mathbf{x})$ is bounded, FA preserves the expressiveness of the underlying backbone, making $\langle \Phi \rangle_{\mathcal{F}}$ maximally expressive even for non-compact groups like $E(n)$.

## F  DETAILS OF ARCHITECTURE

In this section, we provide additional details about the architecture of the Platonic Transformer and the various model configurations used in our experiments. Our framework is designed to be equivariant to roto-translation groups, primarily $SE(n)$ and, through specific configurations, the full Euclidean group $E(n)$.

We denote the core embedding dimension per group element as $d_{\mathrm{hidden}}$. Since our features are functions on a group $\mathcal{G}$ of order $|\mathcal{G}|$, the total feature dimension of a layer is $d_{\mathrm{model}} = |\mathcal{G}| \times d_{\mathrm{hidden}}$. The specific group is determined by the `solid_name` parameter.

For the initial feature processing, input scalars and vectors are first embedded and then lifted into a group-equivariant feature space using an initial lifting operation. This creates a tensor where the channel dimension is expanded by a factor of $|\mathcal{G}|$. An initial group-equivariant linear layer then projects these lifted features to the model's working dimension, $d_{\mathrm{model}}$. Optionally, an equivariant Absolute Positional Encoding (APE), parameterized by `ape_sigma`, can be added at this stage.

The main body of the network consists of a stack of equivariant transformer blocks. Each block contains two main sub-modules: a group-equivariant interaction layer and a feed-forward network (FFN), connected with residual connections. Normalization is applied either before each sub-module or after.

For the group-equivariant interaction layer, we denote the number of attention heads per group element as $n_{\text{head}}$. The total number of effective parallel heads is therefore $|\mathcal{G}| \times n_{\text{head}}$. The dimension of each head, $d_{\text{head}}$, is calculated as $d_{\text{hidden}}/n_{\text{head}}$. The input features are first projected to query, key, and value representations using group-equivariant linear layers. To encode relative spatial information, group-equivariant Rotary Position Embeddings, parameterized by `rope_sigma` and `learned_freqs`, are applied to the query and key vectors. The interaction can then be performed either as a full softmax-based attention mechanism or as a linear-time dynamic group convolution via the `attention` flag. For the Feed Forward Networks (FFNs), we denote the hidden feature dimension as $d_{\text{ffn}} = d_{\text{model}} \times f_{\text{factor}}$. The FFN consists of two group-equivariant linear layers with a GELU activation function in between.

For the final output, two separate readout heads project the features to the desired scalar and vector output dimensions. For graph-level tasks, a pooling operation performs a mean aggregation over the node and group dimensions to produce a final invariant prediction. For node-level tasks, an averaging operation over the group axis projects the features back to standard invariant scalar and equivariant vector representations. Following standard Transformer practices, we apply dropout to the attention weights and FFN activations, and stochastic depth to the outputs of the equivariant transformer blocks.

Particular values for all the important hyperparameters used for the experiments are in the Table.8

## G   HYPERPARAMETER TUNING AND MODEL SELECTION STRATEGY

This section outlines the full procedure used to configure and train our models.

**Baseline Optimization**    To establish a fair point of comparison, we first optimized the general training protocol using only the translation-only equivariant ($T(n)$) models. This initial phase involved tuning the optimizer, learning rate schedule, weight decay, and data augmentations to ensure the baseline models were as competitive as possible. This fixed protocol was then used for all subsequent experiments.

**Hyperparameter Sweep for Model Selection**    With the training protocol fixed, we performed an extensive hyperparameter sweep for both $SE(n)$ and $T(n)$ model classes. This sweep was designed to find the optimal architectural parameters while maintaining an equal computational budget between model families. The parameters and their swept values are summarized in Table 6.

Table 6: Hyperparameter Sweep Configurations.

| Parameter | Configuration | Values |
|---|---|---|
| Hidden Dim | - | $[384, 576, 768, 1152]$ |
| Number of Heads | $T(n)$ **model** (HS=16) | $[24, 36, 48, 72]$ |
| | $T(n)$ **model** (HS=32) | $[12, 18, 24, 36]$ |
| | $SE(n)$ **model** (HS=16) | $[2, 3, 4, 6]$ heads per group element |
| | $SE(n)$ **model** (HS=32) | $[1, 2, 3]$ heads per group element |
| Rope Sigma ($\sigma_{\text{rope}}$) | RoPE frequency scaling | $[0.5 - 2.0]$ |
| Attention | Full Attention / Linear Conv | $[\text{True}, \text{False}]$ |
| Solid Group ($\mathcal{G}$) | Symmetry group | $[\text{Octahedron}, \text{Tetrahedron}, C_{2-8}, D_{4-8}]$ |
| Lambda F ($\lambda_F$) | OMol Force loss weight | $[1.0 - 25.0]$ |
| Batch Size | Samples per batch | $[64 - 512]$ |
| Weight Decay | | $[1e^{-3} - 1e^{-7}]$ |

The hyperparameter sweep was conducted across multiple layers and three random seeds for a *moderate number of epochs* to efficiently explore the configuration space. It should be noted that some configurations are not applicable for the Octahedral group, as its 24 symmetry elements require a minimum of 24 total effective heads (i.e., at least one head per group element). After identifying the

best-performing hyperparameters for both the $SE(n)$ and $T(n)$ model families from this sweep, we proceeded to a final, full-length training run. These selected models were trained for a *large number of epochs* to ensure convergence again with fixed compute budget, producing the final results reported in the main paper.

## H    DETAILS OF EXPERIMENTS ON CIFAR10

### H.1    DESCRIPTION OF THE DATASET

The CIFAR-10 dataset (Krizhevsky, 2009) is a standard benchmark for image classification, consisting of 60,000 32x32 color images across 10 classes. The dataset is divided into a training set of 50,000 images and a test set of 10,000 images.

### H.2    TRAINING DETAILS

For the CIFAR-10 classification task, our experimental setup is closely adapted from the supervised training recipe for Vision Transformers presented in DeiT-III (Touvron et al., 2022). We tokenize each image into a sequence of non-overlapping patches using a patch size of $4 \times 4$ pixels, a key deviation from the ImageNet configurations to suit the lower resolution of the dataset.

The model is trained using the LAMB optimizer, which is subject to a cosine decay schedule following a 5-epoch warm-up period. A comprehensive suite of regularization techniques is employed, including a weight decay of 0.02, Mixup with an alpha value of 0.8, and CutMix with an alpha of 1.0, in addition to model-size-dependent Stochastic Depth. The data augmentation pipeline is built upon the '3-Augment' strategy, incorporating standard Random Resized Crop (RRC), horizontal flips, ColorJitter with a factor of 0.3, and a single, randomly selected transformation from a pool of three: Grayscale, Solarization, or Gaussian Blur.

The training objective is optimized using a Binary Cross-Entropy (BCE) loss, and positional information is supplied to the transformer blocks through a combination of both Absolute Positional Encodings (APE) and Rotary Position Embeddings (RoPE). Further hyperparameter details are available in Table 8.

## I    DETAILS OF EXPERIMENTS ON SCANOBJECTNN

### I.1    DESCRIPTION OF THE DATASET

ScanObjectNN(Uy et al., 2019) dataset is a real-world 3D point cloud dataset. It contains 15,000 objects divided into 15 categories with 2902 unique object instances. It contains background, parts missing, and object deformation elements, which makes the classification task a challenge. The dataset consists of three variants OBJ_BG, OBJ_ONLY and PB_T50_RS, for now the latter is only examined.

### I.2    TRAINING DETAILS

In order to prepare the input point cloud $\mathbf{P} \in \mathbb{R}^{N \times 3}$ for processing by the Platonic Transformer, we follow a preprocessing procedure. Similar to established methods Pang et al. (2023); Yu et al. (2022), we first use Farthest Point Sampling (FPS) to select a set of L=2048 central points, denoted as $\mathbf{P}_C \in \mathbb{R}^{L \times 3}$ with $L = 2048$. Subsequently, for each central point $P_C^i$, we define a local patch $x_p^i \in \mathbb{R}^{K \times 3}$ by identifying its K-Nearest Neighbors (KNN) within the original point cloud P. These local patches serve as the primary input vectors to the Platonic Transformer.

Additionally, to account for the axis-aligned nature of the dataset and to provide the model with a global reference frame, we incorporate rotation augmentation. For each input vector, a rotation matrix is applied. This matrix is either a random rotation or the 3×3 identity matrix, which is concatenated with the input vector to provide the model with information about the global orientation.

## J  DETAILS OF EXPERIMENTS ON QM9

### J.1  DESCRIPTION OF THE DATASET

The QM9 dataset (Ramakrishnan et al., 2014) contains up to 9 heavy atoms and 29 atoms, including hydrogens. We use the train/val/test partitions introduced in Gilmer et al. (2017), which consist of 100K/18K/13K samples, respectively, for each partition.

### J.2  TRAINING DETAILS FOR THE REGRESSION EXPERIMENT

For the QM9 regression task, we train the Platonic Transformer to predict molecular properties. Before being fed to the model, the input molecular geometries are centered by subtracting the mean coordinate of each molecule. To stabilize training, we normalize the target property values by subtracting their mean and dividing by their standard deviation, with these statistics computed over the training set. We employ data augmentation in the form of random $SO(3)$ rotations applied to the coordinates during training.

The model is trained for a total of 1000 epochs using a batch size of 96. We utilize the Adam optimizer with a learning rate of $5 \times 10^{-4}$ and a weight decay of $10^{-8}$. A cosine annealing schedule with a 10-epoch linear warmup adjusts the learning rate throughout training. To prevent exploding gradients, we apply gradient clipping with a maximum norm of 0.5. The training objective is the Mean Absolute Error (MAE) on the normalized target values, while validation and testing are performed by calculating the MAE on the original, unnormalized scale. Our experiments explore different Platonic Transformer configurations, specifically by varying the symmetry group among `trivial_3`, `tetrahedron`, and `octahedron`. Further hyperparameter details are available in Table 8.

We choose FAFormer (Huang et al., 2024) to compare the contribution of Frame Averaging (with a standard Transformer backbone) with our end-to-end trained Platonic Transformer with different platonic solid symmetry groups. Recent efforts in long-context sequence modeling have also pit Transformer-based methods with state-space methods. To provide a similar analysis here, we choose G-Hyena (Moskalev et al., 2025), a recent state-space model that relies on long convolutions for 3D molecular point clouds. We hyperparameter-tune FAFormer[4] and G-Hyena to have similar parameter counts and representational capacity as our best-performing Platonic Transformer on QM9. The baselines are also trained over 500 epochs with a batch size of 96 on a single H200 GPU.

### J.3  ADDITIONAL DETAILS ON WALL-CLOCK TIMINGS

To produce the wall-clock timing for the standard Transformer in Table 4, node features from $B = 64$ QM9 molecules were projected from $d_{\text{in}} = 11$ to $d_{\text{model}} = 512$ using a Linear layer and fed as tokens into a `TransformerEncoderLayer` module provided by PyTorch with 16 heads. We measure wall-clock timings for a forward pass over 10 batches on a single H200 GPU. This offers a reference timing to compare the inference speed of the Platonic Transformer and other geometric baselines like G-Hyena (Moskalev et al., 2025) and the Tensor Field Network (Thomas et al., 2018).

### J.4  EXTENDED EXPERIMENTS ON QM9

We extend our evaluation on the QM9 dataset to include all property targets. Table 3 summarizes these results. We observe that the relative performance hierarchy remains consistent with our main findings: models with higher-order symmetry groups (Octahedron) generally outperform those with lower symmetry (Tetrahedron) and the non-equivariant baseline (Trivial). This confirms that the benefits of the Platonic Transformer's geometric inductive bias generalize across different properties. Crucially, unlike baselines such as EquiformerV2 which rely on target-specific hyperparameter tuning, we employ a single fixed set of hyperparameters across all targets. Despite this constraint, the Platonic Transformer achieves competitive results, suggesting that further performance gains could be realized with target-specific optimization.

---

[4]We use the open-source implementation of FAFormer provided at `https://github.com/Graph-and-Geometric-Learning/Frame-Averaging-Transformer`. We obtain the source code for G-Hyena through private correspondence with the authors.

Table 7: Mean absolute error results on QM9 test set. † denotes using different data partitions. Missing entries will be completed before the camera ready.

| Model | Task Units | $\alpha$ $a_0^3$ | $\Delta\varepsilon$ meV | $\varepsilon_{\text{HOMO}}$ meV | $\varepsilon_{\text{LUMO}}$ meV | $\mu$ D | $C_\nu$ cal/mol K | $G$ meV | $H$ meV | $R^2$ $a_0^2$ | $U$ meV | $U_0$ meV | ZPVE meV |
|---|---|---|---|---|---|---|---|---|---|---|---|---|---|
| DimeNet++ (Gasteiger et al., 2020) | | .044 | 33 | 25 | 20 | .030 | .023 | 8 | 7 | .331 | 6 | 6 | 1.21 |
| EGNN (Satorras et al., 2021)† | | .071 | 48 | 29 | 25 | .029 | .031 | 12 | 12 | .106 | 12 | 11 | 1.55 |
| PaiNN (Schütt et al., 2021) | | .045 | 46 | 28 | 20 | .012 | .024 | **7.35** | **5.98** | .066 | **5.83** | **5.85** | 1.28 |
| TorchMD-NET (Thölke & Fabritiis, 2022) | | .059 | 36 | 20 | 18 | .011 | .026 | 7.62 | 6.16 | **.033** | 6.38 | 6.15 | 1.84 |
| SphereNet (Liu et al., 2022) | | .046 | 32 | 23 | 18 | .026 | **.021** | 8 | 6 | .292 | 7 | 6 | **1.12** |
| SEGNN (Brandstetter et al., 2022)† | | .060 | 42 | 24 | 21 | .023 | .031 | 15 | 16 | .660 | 13 | 15 | 1.62 |
| EQGAT (Le et al., 2022) | | .053 | 32 | 20 | 16 | .011 | .024 | 23 | 24 | .382 | 25 | 25 | 2.00 |
| Equiformer (Liao & Smidt, 2023) | | .046 | 30 | 15 | 14 | .011 | .023 | 7.63 | 6.63 | .251 | 6.74 | 6.59 | 1.26 |
| EquiformerV2 (Liao et al., 2024) | | .050 | **29** | **14** | **13** | **.010** | .023 | 7.57 | 6.22 | .186 | 6.49 | 6.17 | 1.47 |
| PⓇNITA (Bekkers et al., 2024) | | **.038** | 30.4 | 16.0 | 14.5 | .012 | .024 | 8.63 | 8.04 | .235 | 8.67 | 8.31 | 1.29 |
| Platonic Transformer (Trivial, Attn) | | .064 | $45.9_{\pm0.18}$ | $29.4_{\pm0.50}$ | $24.4_{\pm0.35}$ | .028 | - | - | - | - | - | - | - |
| Platonic Transformer (Trivial, Conv) | | .061 | $43.8_{\pm0.65}$ | $26.6_{\pm0.03}$ | $24.0_{\pm0.29}$ | .030 | $.033_{\pm.0006}$ | - | - | $.256_{\pm.0044}$ | - | - | - |
| Platonic Transformer (Tetra, Attn) | | .049 | - | - | - | .012 | - | - | - | - | - | - | - |
| Platonic Transformer (Tetra, Conv) | | .047 | - | - | - | .014 | - | - | - | - | - | - | - |
| Platonic Transformer (Octa, Attn) | | $.049_{\pm.0007}$ | $37.4_{\pm1.36}$ | $22.2_{\pm1.21}$ | $16.7_{\pm0.42}$ | $.010_{\pm.0002}$ | $.024_{\pm.0001}$ | $12.0_{\pm1.00}$ | $12.0_{\pm0.26}$ | $.222_{\pm.0062}$ | $11.9_{\pm1.73}$ | $13.0_{\pm0.00}$ | $1.3_{\pm0.01}$ |
| Platonic Transformer (Octa, Conv) | | $.048_{\pm.0013}$ | $33.8_{\pm1.00}$ | $17.7_{\pm0.51}$ | $15.7_{\pm0.25}$ | $.012_{\pm.0001}$ | $.026_{\pm.0001}$ | $11.0_{\pm0.45}$ | $11.7_{\pm0.20}$ | $.184_{\pm.0110}$ | $13.9_{\pm0.85}$ | $10.9_{\pm0.36}$ | $1.4_{\pm0.03}$ |

Table 8: Hyperparameter for all datasets

| Hyperparameter | QM9 | OMol25 | CIFAR10 | ScanNetNN |
|---|---|---|---|---|
| **Architecture** | | | | |
| hidden_dim | 1152 | 1152 | 768 | 576 |
| layers | 14 | 14 | 12 | 12 |
| num_heads | 72 | 72 | 12 | 12 |
| **Positional encoding** | | | | |
| freq_sigma | 0.5 | 0.5 | 1 | 18 |
| ape_sigma | 0.5 | None | 10 | 10 |
| learned_freqs | True | True | True | True |
| **Attention / readout** | | | | |
| dropout | 0.0 | 0.0 | 0.0 | 0.1 |
| drop_path_rate | 0.0 | 0.0 | 0.1 | 0.0 |
| mean_aggregation | False | False | False | False |
| **Training** | | | | |
| lr | 5e-4 | 2e-4 | 8e-4 | 8e-4 |
| batch_size | 96 | 64 | 256 | 128 |
| epochs | 1000 | 22 | 500 | 500 |
| warmup | 10 | 5 | 20 | 10 |
| weight_decay | 1e-8 | 1e-6 | 0.05 | 1e-6 |
| lambda_F | - | 12.0 | - | - |
| cosine_scheduler | True | True | True | True |
| precision | 32 | 32 | 32 | 32 |
| gpus | 1 | 4 | 1 | 1 |

# K  DETAILS OF EXPERIMENTS ON OMOL25

## K.1  DESCRIPTION OF THE DATASET

For large-scale molecular experiments, we use the Open Molecules 2025 (OMol25) dataset (Levine et al., 2025), a comprehensive collection of over 100 million Density Functional Theory (DFT) calculations performed at the wB97M-V/def2-TZVPD level of theory. This dataset is notable for its vast chemical and structural diversity, encompassing 83 elements and systems up to 350 atoms. The structures are drawn from a wide range of chemical domains, including small molecules, biomolecules, metal complexes, and electrolytes, and feature varied charges, spin states, conformers, and reactive geometries.

The OMol25 dataset is organized into several training sets and splits for validation and testing to ensure consistent and robust model evaluation. The full training set, "All," contains over 100 million DFT calculations. For more computationally efficient training and development, a smaller, uniformly sampled "4M" split is provided, containing approximately 4 million structures. Our work primarily utilizes the "Neutral" split, which consists of approximately 34 million charge-neutral, singlet structures drawn from established community datasets like ANI-2X, GEOM, and SPICE2. This split is designed to benchmark model performance on familiar organic chemistry space without the added complexity of variable charge and spin.

For validation and testing, OMol25 provides several out-of-distribution (OOD) splits designed to evaluate model generalizability. The primary validation set ("Val Comp") consists of structures with compositions held out from the training set. Further specialized test sets include held-out organic and metal-complex reactions ("Test Reactivity"), experimental crystal structures from the Crystallography Open Database ("Test COD"), and unique anion structures ("Test Anions"), among others. The core task is Structure to Energy and Forces (S2EF), where models are evaluated on their ability to predict the total energy of a structure and the per-atom forces, with Mean Absolute Error (MAE) being the primary metric.

## K.2 TRAINING DETAILS

We partition the "Neutral" split of the OMol25 dataset into an 80% training set and a 20% validation set, and report the final results on the official test set. The model is trained using an AdamW optimizer with a learning rate of $2 \times 10^{-4}$ and a weight decay of $10^{-6}$. The learning rate is managed by a cosine decay schedule, which includes a linear warmup period over the first 5 epochs. Training is conducted for a total of 22 epochs using a batch size of 64.

The training objective is a weighted sum of two components: a **Mean Squared Error (MSE)** loss for the total energy and a **Mean Absolute Error (MAE) on the force vectors**. The force loss is calculated as the average L2 norm (Euclidean distance) of the error between the predicted and target force vectors for each atom. To balance these two targets, the force loss component is weighted by a factor of $\lambda_F = 12.0$.

To ensure stable training on this large-scale task, we normalize the target energies. We apply a linear referencing scheme to the raw DFT energies. This method normalizes the total energy by subtracting the pre-computed DFT energies of the constituent isolated atoms:

$$E_{\text{ref}} = E_{\text{DFT}} - \sum_{i=1}^{N} E_i^{\text{atom}} \tag{29}$$

where $E_{\text{ref}}$ is the target value for the model, $E_{\text{DFT}}$ is the system's total energy, $N$ is the number of atoms, and $E_i^{\text{atom}}$ is the pre-computed DFT energy of an isolated atom of the same species as atom $i$. This procedure is consistent with the methodology used for the OC22 dataset (Tran et al., 2023) and helps maintain comparability with other large-scale models.

Given the significant computational requirements for training on OMol25, we established a fixed compute budget to ensure a fair comparison between models. Each model was trained for a maximum duration of 5 days on a node equipped with 4x NVIDIA 6000Ada GPUs. The detailed hyperparameters for our model configuration on this dataset are summarized in Table 8.

# L EXPERIMENTS ON LEARNED KEY PROJECTIONS

In Section 4.1 and Remark 1 of the main text, we motivate our design choice of using fixed key vectors ($k_j = 1$)—as opposed to learned linear projections ($k_j = W^K f_j$)—by citing training instabilities observed on molecular datasets. In this section, we report an empirical analysis of this observation.

## L.1 INSTABILITY OF LEARNED KEYS

To investigate the impact of learned keys, we conducted experiments on the QM9 dataset using the standard hyperparameters defined in Appendix J. We compared the standard model (fixed keys) against a variant with learned key projections. We performed this comparison for both the full

Attention mechanism and the linear Convolutional variant, training for 300 epochs across two random seeds.

The results are illustrated in Figure 4. As shown in Figure 4a, when using the full Attention mechanism, the introduction of learned keys ('use_key=True') leads to severe training instability. Both runs utilizing learned keys exhibit divergence around epoch 10, with one run failing to complete. In contrast, the fixed key formulation ('use_key=False') trains smoothly.

In the linear Convolutional mode (Figure 4b), training remains stable for both configurations. However, as shown in Figure 4c, the learned keys provide no performance benefit; in fact, the model with fixed keys achieves a lower Test MAE. This suggests that even when stability is maintained, the entanglement of content and geometry introduced by learned keys does not improve generalization for this physical task.

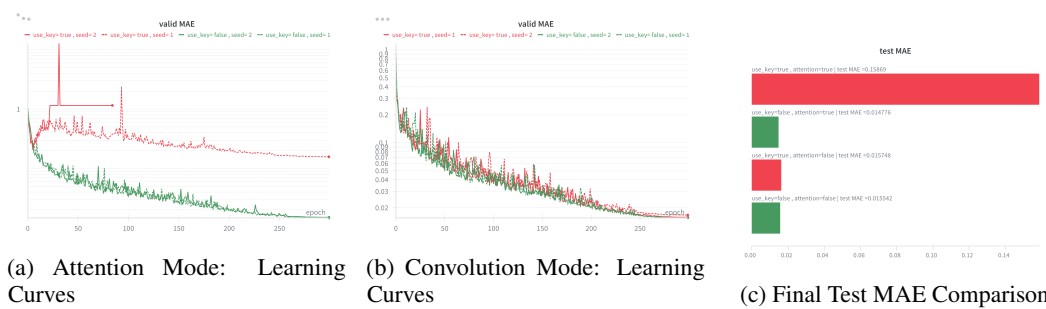

(a) Attention Mode: Learning Curves
(b) Convolution Mode: Learning Curves
(c) Final Test MAE Comparison

Figure 4: Impact of Learned Key Projections on Stability and Performance. (a) When using full attention, learned keys cause rapid divergence/instability around epoch 10. (b) In convolutional mode, training is stable, but (c) fixed keys consistently outperform learned keys in final accuracy.

### L.2 MITIGATING INSTABILITY VIA REGULARIZATION

We further hypothesized that the instability in the Attention setting might be mitigated by stronger regularization. We performed a sweep of weight decay values ranging from $10^{-1}$ to $10^{-8}$ for the model with learned keys.

Figure 5 presents these results. Figure 5a shows that while high weight decay values ($10^{-1}$ to $10^{-4}$) can stabilize the training, reducing the weight decay below $10^{-4}$ immediately reintroduces the instability observed in the previous experiment. Figure 5b shows that the the weight decay should be as small as possible while still leading to stable training, however, still the best performance is lagging considerably behind the results of our default weight decay setting of $10^{-8}$ for the constant key scenario (Figure 4).

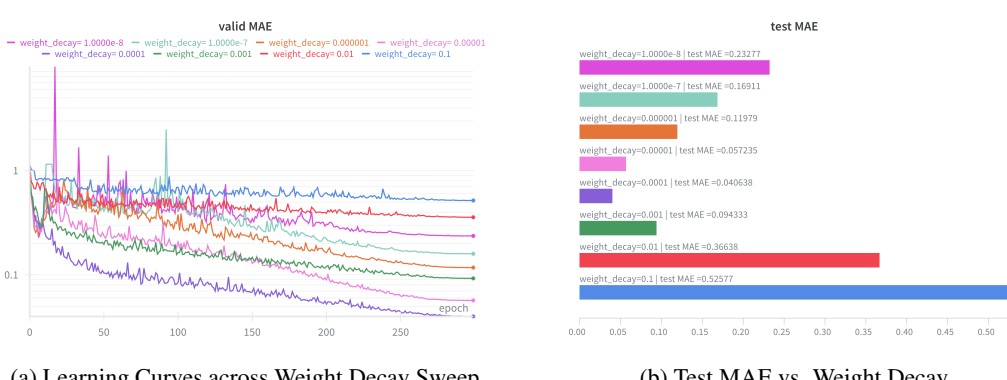

(a) Learning Curves across Weight Decay Sweep
(b) Test MAE vs. Weight Decay

Figure 5: Can Weight Decay Fix Learned Keys? (a) Strong weight decay stabilizes training, while values $< 10^{-4}$ lead to divergence. (b) The final test MAEs of each model.

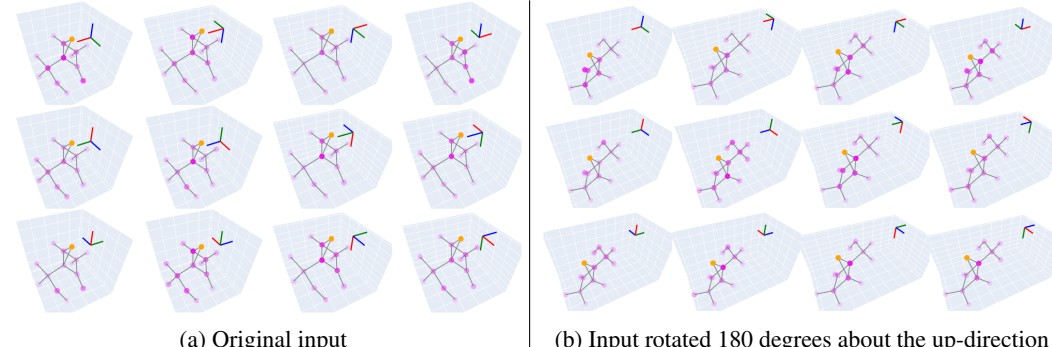

(a) Original input | (b) Input rotated 180 degrees about the up-direction

Figure 6: We visualize the attention score between the orange node and all others, where an increased color intensity indicates an increased attention score. The subplots correspond to 12 different frames in the same head of an octahedral Platonic Transformer layer (there are 12 more frames not visualized here). The attention is broadly focused on locality but with distinct directional biases. The equivariance of the model can be observed by comparing the attention scores in the sub-figures. For instance, the attention pattern in the top-left frame in Figure 6a is the same as the one in the top-right frame in Figure 6b, but rotated 180 degrees.

**Conclusion:** These experimnets confirm that for physical tasks like QM9, using fixed keys ($k = 1$) is not merely a simplification but an important design choice that ensures training stability and solid performance. While we report results over 300 epochs here (compared to 1000 in the main experiments), the early onset of instability and the consistent performance gap makes it implausible that heavy regularized learned-key platonic transformers could match the constant-key performance.

## M  VISUALIZATIONS OF LEARNED ATTENTION SCORES

To show the directional attention learned in the attention head, we visualize examples over attention patterns in different frames $g \in \mathcal{G}$ in Figure 6.

## N  IMPLEMENTING PLATONIC TRANSFORMERS IN THE FOURIER DOMAIN OF FINITE GROUPS

With increasing hidden dimension (while not increasing sequence length), transformer blocks spend more and more of their total compute time in the pointwise linear layers. To improve speed it can then be worthwhile to implement the pointwise equivariant linear layers in the Fourier domain of the rotation group, a technique that has recently been successfully employed in computer vision (Bökman et al., 2025; Nordström et al., 2025). Considering the Fourier domain also sheds light on the connections between Platonic Transformers and equivariant networks with general steerable feature spaces (Cesa et al., 2022).

In this section we demonstrate how a Fourier domain implementation can improve computational efficiency in Platonic Transformers. In the Fourier domain, equivariant linear layers are block-diagonal, drastically reducing the required number of FLOPs for both forward and backward passes. We will see that with the number of hidden dimensions considered in this paper, a naive PyTorch implementation is not efficient enough to realize the reduction in FLOPs in terms of a substantial reduction in training throughput, but at a moderately higher number of hidden dimensions, there are throughput gains. This suggests that future scaling of Platonic Transformers will benefit from being implemented in the Fourier domain, and that more efficient implementations than our current one would be able to improve throughput even at smaller number of hidden dimensions.

We will use the tetrahedral symmetry group as a running example in this section. The reader is cautioned that the representations discussed in this section are representations of the rotation group, in contrast to the representations of the translation group discussed in Appendix A.

## N.1 INTRODUCTION TO THE FOURIER THEORY OF FINITE GROUPS

The representation theory of finite groups is a well studied topic with many good text books. We recommend (Serre, 1977) for more detailed background than given here. Note that we consider vector spaces over the real numbers, which leads to a slightly more involved representation theory than complex numbers, see (Serre, 1977, Section II.12).

Recall from Appendix A.1 that a representation of a group $\mathcal{G}$ is a group homomorphism $\rho : \mathcal{G} \to GL(V)$, where $V$ is a vector space. We will here consider finite real vector spaces $V = \mathbb{R}^n$ so that $\rho(g)$ can be considered real-valued invertible matrices. An irreducible representation is one where the matrices $\{\rho(g)\}_{g \in \mathcal{G}}$ can not be simultaneously block-diagonalized. Any finite group $\mathcal{G}$ has a finite number (up to ismorphisms) of irreducible representations (irreps) $\{\rho_i\}$ and they can be computed given the multiplication table of the group. Irreps are important because we can decompose any finite representation $\rho$ into a direct sum of irreps by performing a change of basis, so statements about general representations often reduce to statements about irreps.

The features in Platonic Transformers are functions from $\mathcal{G}$ to $\mathbb{R}^C$, that transform under the left regular representation as explained in Appendix B. In order words, the representation that acts on them is a direct sum of $C$ copies of the regular representation of $\mathcal{G}$. Let this representation be denoted $\tilde{\rho}$. Decomposing $\tilde{\rho}$ into irreps, we obtain

$$\tilde{\rho}(g) = Q \left( \bigoplus_i \rho_i(g)^{\oplus m_i} \right) Q^{-1} \tag{30}$$

for some multiplicities $m_i$ of each irrep and a change of basis matrix $Q$ that can be taken to be orthogonal.

Now, Schur's lemma says that any equivariant linear map between non-isomorphic irreps $\rho_i \neq \rho_j$ must be constant zero. Further, the space of equivariant linear maps between $\rho_i$ and itself is 1-, 2-, or 4-dimensional and isomorphic (as a division algebra over $\mathbb{R}$) to the real numbers, complex numbers, or quaternions depending on whether $\rho_i$ is of so-called real, complex or quaternion type. (The type of $\rho_i$ can be computed.) This means that any linear map that is equivariant from $\tilde{\rho}$ to $\tilde{\rho}$ is actually block diagonal after having performed the change of basis in (30), in particular so are the group convolutions used in Platonic Transformers.

For cyclic groups, the block-diagonalization corresponds to the fact that convolutions are pointwise multiplications in the Fourier domain[5].

## N.2 FOURIER THEORY OF THE TETRAHEDRAL GROUP

Let us now consider the Tetrahedral rotation group as $\mathcal{G}$, consisting of the twelve rotational symmetries of a regular tetrahedron. This group is isomorphic to the alternating group $A_4$ and has three real irreps. The real irreps of the tetrahedral group are given by the one-dimensional trivial representation

$$\rho_1(R) = 1, \tag{31}$$

the three-dimensional standard representation

$$\rho_3(R) = R \tag{32}$$

and a two-dimensional representation $\rho_2$ that is defined as follows. Note that any element in $\mathcal{G}$ is either the identity, a rotation by $2\pi/3$ radians (there are 8 of these) or a rotation by $\pi$ radians (there are 3 of these). For the identity and rotations by $\pi$,

$$\rho_2(R) = \begin{pmatrix} 1 & 0 \\ 0 & 1 \end{pmatrix}. \tag{33}$$

The rotations by $2\pi/3$ fall into two conjugacy classes of four elements each, where one conjugacy class contains the inverses of the second. We can arbitrarily choose one of the conjugacy classes and define

$$\rho_2(R) = \begin{pmatrix} \cos(2\pi/3) & -\sin(2\pi/3) \\ \sin(2\pi/3) & \cos(2\pi/3) \end{pmatrix} \tag{34}$$

---

[5]This requires working over the complex numbers, over the real numbers the pointwise multiplications turn into $2 \times 2$ matrix multiplications, again a block-diagonal structure.

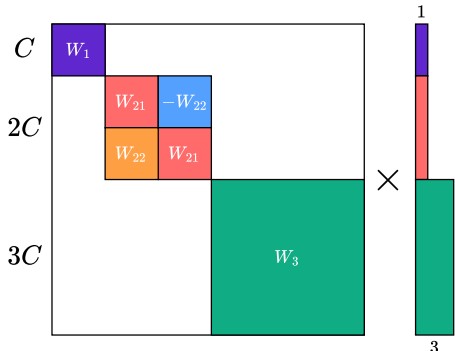 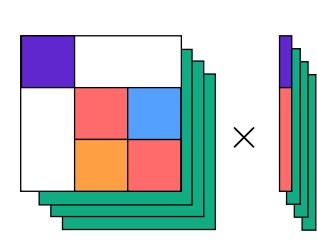

(a) The block-diagonal structure of an equivariant weight matrix in the Fourier domain.

(b) We can implement the linear layer as a batched matrix-vector multiplication with four batches.

Figure 7: We visualize the weight matrices for linear layers that are equivariant under the tetrahedral rotation group, implemented in the Fourier domain. Each subfigure shows weights to the left and features to the right. Purple features transform according to $\rho_1$ (or technically $\rho_1 \otimes I_C$ since there are $C$ copies of $\rho_1$), red features according to $\rho_2$ (by multiplication by $\rho_2(g) \otimes I_C$ from the left) and green features according to $\rho_3$ (by multiplication by $\rho_3(g)^\top$ from the right (if we flattened the green features, they would transform by $\rho_3(g) \otimes I_{3C}$ from the left)). The weight matrix is parameterized by the $C \times C$ matrices $W_1, W_{21}, W_{22}$ and the $3C \times 3C$ matrix $W_3$, yielding a total of $12C^2$ learnable parameters. The total number of multiplications to compute the linear layer implemented as a batched matrix-multiplication in 7b is $4 \cdot (3C)^2 = 36C^2$, yielding a $4\times$ FLOP reduction versus an ordinary layer from $12C$ to $12C$ dimensions ($144C^2$ multiplications).

there, which implicitly defines the values for the second conjugacy class to be the inverse of the above.

It can be computed that $\rho_1$ and $\rho_3$ are both of real type, while $\rho_2$ is of complex type. Hence, equivariant linear maps from $\rho_1$ to $\rho_1$ are parameterized by one value, and the same for $\rho_3$. Equivariant maps from $\rho_2$ to $\rho_2$ are instead parameterized by two values (this is because $\rho_2$ splits into two irreps over the complex numbers).

It can also be computed (or recovered from general facts of the Fourier transform over finite groups) that the representation $\tilde{\rho}$ acting on features with $C$ channels in a tetrahedral Platonic Transformer splits into $C$ copies of $\rho_1$, $C$ copies of $\rho_2$ and $3C$ copies of $\rho_3$ (as a sanity check, we recover all $C + C \cdot 2 + 3C \cdot 3 = 12C$ dimensions).

As mentioned, Schur's lemma now implies that equivariant linear maps from $\tilde{\rho}$ to itself are block-diagonal. The map from copies of $\rho_1$ to copies of $\rho_1$ is parameterized by a $C \times C$ matrix, the map from copies of $\rho_2$ to copies of $\rho_2$ is parameterized by two $C \times C$ matrices (because $\rho_2$ is of complex type) and the map from copies of $\rho_3$ to copies of $\rho_3$ is parameterized by a $3C \times 3C$ matrix. Again, a sanity check gives that the full equivariant layer is then parameterized by $C^2 + 2C^2 + (3C)^2 = 12C^2$ values, which is the same as the group convolution discussed in Section 3.3.

We visualize the weight structure in Figure 7a.

### N.3 Implementation

We implement a version of the Platonic Transformer with tetrahedral equivariance and all linear layers (i.e. in the MLP and projections in multi-head attention) in the Fourier domain. We transform back to the spatial domain at each non-linearity and at the RoPE-attention layers and to the Fourier domain after these layers. This transforming back-and-forth incurs an overhead that goes to zero as the hidden dimension increases (since it is just the $12 \times 12$ matrix $Q$ applied to each channel $C$), however it is non-negligible at low–medium number of hidden dimensions, because it involves non-contiguous reshapes.

The maximum FLOP saving that can be obtained from changing a linear layer to be in the Fourier domain is going from $(12C)^2 = 144C^2$ operations to $C^2 + (2C)^2 + 3 \cdot (3C)^2 = 32C^2$, i.e. a saving of $4.5$ times. However, in order to make the implementation more efficient in pure PyTorch, we opt to implement the mappings for $\rho_1$ and $\rho_2$ as one single $3C \times 3C$ matrix, enabling the whole linear layer to be implemented as a batched matrix multiplication with four $3C \times 3C$ weight matrices, as illustrated in Figure 7b. This batched implementation uses $4 \cdot (3C)^2 = 36C^2$ operations, yielding a maximum potential compute saving of $4$ times.

### N.4 Throughput Benchmarking

We benchmark the training time per epoch on a subset of 20k molecules on the OMol25 task, using PyTorch's `torch.compile`. These timing runs are on a single NVIDIA RTX6000 GPU. We keep all hyperparameters constant as in the main experiments, except for varying the number of hidden dimensions. The results are presented in Table 9. It is clear that as we increase the number of hidden dimensions, a Fourier implementation starts paying off more and more. Notably, since the standard spatial implementation is equal to non-equivariant Transformers in computational cost, the efficiency improvement of the Fourier implementation is a benefit of equivariant architectures over non-equivariant ones. We emphasize that our Fourier implementation is not well-optimized, so further throughput improvements should be available.

Table 9: Training times per epoch (seconds) on a subset of OMol25 with 20k examples. We compare a tetrahedral Platonic Transformer implemented in the spatial domain with one implemented in the Fourier domain.

| Implementation | Hidden dimension | | | | | |
| --- | --- | --- | --- | --- | --- | --- |
| | 576 | 864 | 1152 | 1440 | 1728 | 2016 |
| Spatial (standard) | 18 | 23 | 29 | 40 | 49 | 63 |
| Fourier | 19 | 22 | 27 | 32 | 38 | 45 |

## O  Disclosure of LLM Usage

We declare that the use of LLMs for writing this paper was limited to general-purpose writing assistance. Specifically, we used them only to polish the wording of text sections and in no way to generate the research ideas or technical results and proofs presented in this paper.

