# OpenReview forum: "Platonic Transformers: A Solid Choice for Equivariance"
_ICLR.cc/2026/Conference — Submitted to ICLR 2026_

### Official Review · Reviewer_uQvE · 2025-10-27

**Soundness:** 3
**Presentation:** 3
**Contribution:** 3
**Rating:** 4
**Confidence:** 4

**Summary:**

This paper introduces considers the problem of designing transformer architectures for points sets, which are equivariant to rotations and translations. Previous work has accomplished this by using novel equivariant transformers which are more computationally intensive than standard (non-equivariant) transformers. The method suggested in this paper relaxes the problem to equivariant to a discrete subgroup of the rotation group,  and using this relaxation shows how standard transformers can be modified to accomplish equivariance without compromising efficiency.

**Strengths:**

Reading this paper was a pleasant experience and the mathematics seems correct. The basic idea of the paper makes sense to me: building an efficient equivariant transformer built on the basis of existing transformers. The empirical results on QM9 especially give some basic corroboration of this idea: comparable accuracy with far superior inference time.

**Weaknesses:**

1. The extent of the empirical results is not very convincing:
* In QM9, only two of the targets are reported. A skeptic could suspect that these are the only targets on which the method performs well. Results on all targets should be reported.
* In OMol25, I understand the point you make in the first table. Still, it would be more convincing if you could also compete with the reported results from the Esen  paper. Take all the training time you like.
2. I have some point on the writing I will discuss in the questions section. These can be addressed in a small revision for the camera ready version.

I will increase my score if these issues are resolved.

**Questions:**

Regarding your discussion of frame averaging in line 424: is there any essential reason why one would require a separate forward pass for each frame element? Couldn't you, as a preprocessing step, turn the input from an n by 3 input to an n by 3 by 4 input (where 4 is the number of SO(3) invariant frames).


**Please do not related to what is below in the rebuttal** these are just suggestions for improvements of writing  for the next version of the paper:
* Line 47: "thereby expanding..." I feel like the logic of this sentence could be improved
* Line 84: the second $k_j$ should be $v_j$
* The explanation in Section 2 is pretty good, but it would be very helpful if you defined $p$ in the beginning, explained what its dimensions are and what translation and rotation invariance would mean for $p$.
* The footnote in page 4 continues to page 5 which is a bit strange.
* While reading the paper I was wondering whether SO(3) equivariance was being relaxed to equivariance over the chosen subgroup. The answer only became apparent to me when this information was disclosed in subsection 5.1. I would appreciate being more forthcoming about this information earlier and more frequently.
* Line 356: I didn't follows why ScanObjectNN is considered a non-equivariant task. Do objects appear their in a predefined orientation?
* Line 402: impact is misspelled
* The grammar in the last sentence in the ethics statement is off.

---

> ### Author Response · Authors · 2025-11-20
>
> We thank the reviewer for their thorough assessment of our work, and contributing to improve the work. Moreover, we are happy to see that the reviewer had a pleasent experience while reading the manuscript.
>
> ### Weaknesses
>
> #### **The extent of the empirical results is not very convincing: [...]**
>
> #### **[All targets for QM9]**
>
>  We do agree that only reporting two targets for QM9 is not very elaborate. Therefore we setup the runs for all other targets and will have the results in around a week from now.
>
> #### **[Compete with Esen on Omol25]**
>
> We understand the point that the reviewer makes, but a fair comparison with the published results from Esen on OMol unfortunately requires computational resources that we do not have.
> We have looked into the smaller 4M subset of OMol, but even there a training run would take longer than the ICLR discussion phase with our current computational budget.
>
>
> ### Questions
>
> #### **Regarding your discussion of frame averaging in line 424: is there any essential reason why one would require a separate forward pass for each frame element? Couldn't you, as a preprocessing step, turn the input from an n by 3 input to an n by 3 by 4 input (where 4 is the number of SO(3) invariant frames).**
>
> We thank the reviewer for this insightful question, which highlights a crucial practical consideration in implementing SO(3)-equivariant models.
> The reviewer is entirely correct that the four frame elements (the basis and its three reflections) can be concatenated as a pre-processing step, effectively transforming the input from an n × 3 representation to an n × 3 × 4 tensor. This approach is mathematically equivalent to four separate forward passes followed by averaging. However, the reason we explicitly discussed the separate forward passes in the paper was to highlight a critical practical trade-off: memory efficiency and the resulting maximum batch-size.
> (a) Memory Constraint: Concatenating the frames into an n×3×4 tensor effectively
> multiplies the memory required for the input data and intermediate activations
> by a factor of 4.
> (b) Batch Size Limitation: This multiplied memory footprint severely limits the
> maximum achievable batch size on a fixed GPU memory budget. As the reviewer
> notes, one would effectively require 4× the memory (or be forced to use a 4×
> smaller batch size).
> In the context of modern deep learning, where the community is rapidly moving
> towards large-scale, multi-modal pretraining paradigms, maintaining a large batch size is often paramount for stable, effective training and achieving state-of-the-art results. Approaches that significantly compromise the maximal batch-size tend to have lower utility for large-scale applications. Our PlatoFormer architecture is specifically designed to be highly compute- and memory-efficient, enabling the use of large batch sizes while maintaining SO(3)- equivariance. Therefore, emphasizing the cost of the frame averaging procedure is important because the practical implementation needs to prioritize memory efficiency to achieve the necessary scale. This design philosophy further supports our central thesis that equivariance and scaling can work together.
>
> #### **Writing suggestions**
>
> Thank you for these detailed suggestions! We will correct the typos, grammar, and improve clarity in the mentioned sections (Lines 47, 84, Section 2, footnote, ScanObjectNN, etc.) in the final version. We especially appreciate the feedback on making the relaxation to a discrete subgroup clearer from the beginning and will ensure this is addressed.

---

> > ### Author Response · Authors · 2025-11-20
> >
> > We thank the reviewer for their constructive feedback and commitment to enhancing our work. We are happy to provide any additional details needed to completely satisfy any lingering concerns. Assuming the presented clarifications have fully addressed the points raised, we respectfully ask the reviewer to consider improving the manuscript's overall grade.

---

> > > ### Comment · Reviewer_uQvE · 2025-11-22
> > >
> > > I would like to thank the authors for the answers. I am happy with the answers regarding omol25 and frames, and am looking forward to the results for the rest of the targets in the QM9 comparison.

---

> > > > ### Author Response · Authors · 2025-12-03
> > > >
> > > > Thank you for acknowledging our efforts!
> > > >
> > > > Finally, as requested we have extended our evaluation to include all 12 QM9 targets. Due to computational constraints, some runs are still completing, but the current results can be found in the new Appendix **J.4 EXTENDED EXPERIMENTS ON QM9**. We ran our model on all targets with 3 seeds each and report the mean and standard deviation.
> > > >
> > > > The preliminary results confirm that the hierarchy $\text{Trivial} < \text{Tetrahedron} < \text{Octahedron}$ holds consistently across targets, which underpins the robustness of our geometric inductive bias. While we achieve state-of-the-art performance on several targets, we acknowledge that on others our results fall within the range of some older baselines, but behind state-of-the-art methods like EquiformerV2 and PONITA. This is expected given that we use a **single fixed set of hyperparameters** for all targets, whereas baselines typically employ target-specific tuning. Finally, we distinguish our evaluation by reporting the mean and standard deviation over 3 independent seeds for each experiment. This multiple seed evalution protocol, omitted in comparable literature where single-run results are common, further highlights the stability of our method and the reliability of the reported performance.
> > > >
> > > > All in all we believe this complete table is a valuable addition to the paper, and thank the reviewer for encouraging us to pursue it.

---

### Official Review · Reviewer_rkTK · 2025-10-31

**Soundness:** 2
**Presentation:** 3
**Contribution:** 3
**Rating:** 4
**Confidence:** 4

**Summary:**

The paper presents an efficient method to build equivariance in the standard Transformer architecture with respect to discrete symmetry group $G \subset SO(3)$.
The method is specifically based on rotary positional embedding (RoPE), where the main idea is to run RoPE-based attention in parallel over a fixed set of group elements (reference frames) with shared weights across the selected frames (frames are sampled from the finite subgroup of $SO(3)$).
The authors evaluate their methods on different input domains including: 2D images (CIFAR-10 dataset), 3D point clouds (ScanObjectNN dataset), and 3D molecules (QM9 and OMol25 datasets).

**Strengths:**

- I think the proposed idea is simple and interesting. With limited modifications to the standard Transformer, the authors can achieve equivariance w.r.t. discrete subgroups from SO(3) in an efficient and effective way.

- The authors also show good evaluations for their proposed method, and that it can be applied to different input domains (images, point clouds, and molecules).

**Weaknesses:**

* The baselines are limited to convolution in the case of vision domains. I think the paper should include stronger models in the literature and works that consider similar symmetry subgroups.
* The proposed method only achieves approximate equivariance w.r.t. $SE(3)$ or equivariance w.r.t. finite subgroups of $SO(3)$, but the paper claims multiple times using the phrase “full equivariance to Euclidean transformations”. I think some rewrites might be required, and the authors should state clearly that the goal of the idea is to build approximate equivariance/equivariance w.r.t. discrete subgroups.
* The propositions need some more illustration, and it would be benefical to mention which symmetry group is considered. For example, Prop. 1 states  “A global roto-reflection $R \in G$ applied to the input point cloud.." Do you mean the discrete subgroup in this case?

**Questions:**

As 3D point clouds and molecules require continuous equivariance to the SO(3) symmetry group, how do the authors deal with this case? Did you apply rotation augmentations during training?

---

> ### Author Response · Authors · 2025-11-20
>
> We thank the reviewer for their thorough assessment of our work, and contributing to improve the work. Moreover, we are glad to see that the reviewer appreciates our simple, efficient and effective approach of incorporating equivariance.
>
> ### Weaknesses
>
> #### **The baselines are limited to convolution in the case of vision domains. I think the paper should include stronger models in the literature and works that consider similar symmetry subgroups.**
>
> We do agree with the reviewer that the manuscript could from benefit more comparisons with existing baselines. We would like to clarify that one of our baselines is, in fact, a standard Vision Transformer (ViT). In our framework, this standard ViT is represented by the "Trivial Group" model (denoted $\emptyset*$ in the Attention column in Table 1). This model uses the same Transformer architecture and RoPE as our other variants but has no rotational symmetry, making it exactly equivalent to a standard, translation-equivariant rope-based ViT. We believe this is the most relevant baseline, as it allows for a direct and fair ablation: it compares a standard ViT directly against our symmetry-aware variants (e.g., $C_4$, $D_4$) within the exact same architectural and computational framework. We have updated the text in Section 5.2 to make this explicit.
>
> #### **The proposed method only achieves approximate equivariance w.r.t. SE(3) or equivariance w.r.t. finite subgroups of SO(3), but the paper claims multiple times using the phrase “full equivariance to Euclidean transformations”. I think some rewrites might be required, and the authors should state clearly that the goal of the idea is to build approximate equivariance/equivariance w.r.t. discrete subgroups.**
>
> Thank you for pointing this out, we do agree this could have been posed clearer. We have edited the manuscript to make it clearer that the equivariance is w.r.t. a finite subgroup of $O(3)$.
>
> #### **The propositions need some more illustration, and it would be benefical to mention which symmetry group is considered. For example, Prop. 1 states “A global roto-reflection applied to the input point cloud.." Do you mean the discrete subgroup in this case?**
>
> Prop. 1 indeed refers to the discrete subgroup $\mathcal{G}$. We have clarified this in the manuscript. For clarification, by *illustration*, do you mean figures or more text explaining the propositions?
>
> ### Questions
>
> ####  **As 3D point clouds and molecules require continuous equivariance to the SO(3) symmetry group, how do the authors deal with this case? Did you apply rotation augmentations during training?**
>
> Yes, we apply random rotation augmentations to our data while training to cover the continuous rotations. Several recent works have posited that full, strict architectural equivariance is not always necessary for SO(3) [A, B, C]. In our experiments we observe that introducing equivariance wrt finite subgroups of SO(3) still helps performance, and crucially this comes at no added computational cost.
>
>
> [A] Joshi et al, All-atom Diffusion Transformers: Unified generative modelling of molecules and materials, ICML 2025
>
> [B] Elhag et al, Relaxed Equivariance via Multitask Learning, LoG 2025
>
> [C] Brehmer et al, Does Equivariance Matter at Scale?, TMLR 2025
>
> We sincerely thank the reviewer once again for their constructive comments. If, after reviewing our responses and modifications, the reviewer feels all concerns have been addressed, we respectfully ask them to consider increasing the final score; otherwise, we remain fully open to addressing any further questions or clarification needs.

---

> > ### Author Response · Authors · 2025-11-20
> >
> > We sincerely appreciate the reviewer's time and effort in helping us refine this manuscript. We welcome any remaining questions and are available for immediate clarification on any points. If the provided responses have successfully resolved all prior concerns, we would be grateful if the reviewer would reassess and consider an increased evaluation score.

---

### Official Review · Reviewer_Pkaq · 2025-11-01

**Soundness:** 3
**Presentation:** 3
**Contribution:** 2
**Rating:** 6
**Confidence:** 3

**Summary:**

The paper presents the Platonic Transformer, which incorporates geometric equivariance into standard Transformers by defining attention relative to Platonic solid symmetry groups. Using a group-theoretic reinterpretation of Rotary Position Embeddings, it achieves equivariance to translations and discrete rotations without changing the Transformer architecture or cost. Experiments across vision, 3D, and molecular tasks show that it matches or exceeds state-of-the-art models while preserving efficiency.

**Strengths:**

By defining attention relative to Platonic solid symmetry groups and reinterpreting Rotary Position Embeddings as dynamic group convolutions, it unites group theory with Transformer attention in a principled way. The work is technically strong, clearly presented, and demonstrates broad empirical effectiveness across image, 3D, and molecular domains.

**Weaknesses:**

1. While the Platonic Transformer effectively maintains architectural flexibility, its design introduces a nontrivial trade-off between group size and model expressivity. To control computational overhead, the authors fix the total feature dimension and proportionally reduce the channel size per group element as $|G|$ increases. Although this strategy preserves overall parameter count and computational cost, it implicitly limits the representational capacity of each frame, potentially weakening the model’s ability to capture rich geometric variations when larger symmetry groups are used. Consequently, there exists a tension between achieving stronger equivariance (by enlarging $|G|$) and maintaining sufficient feature expressivity within each frame. A more thorough analysis or ablation of this trade-off, quantifying how equivariance benefits degrade or saturate as channel capacity per group element decreases, would strengthen the paper’s claims on scalability and generalization.

2. Comparison with stronger baselines: While results are competitive, the paper omits direct comparisons with recent high-performing equivariant Transformers such as SE(3)-Transformer (Fuchs et al., 2020), Euclidean Fast Attention (Frank et al., 2024), and Geometric Algebra Transformers (Brehmer et al., 2023). Including these would contextualize the claimed efficiency-equivariance trade-off more rigorously.

**Questions:**

1. Appendix L suggests potential computational gains from Fourier-domain implementations. Could the authors provide more concrete benchmarks or scaling results demonstrating the actual speedup at different model sizes? How does this compare to standard Transformer throughput?

2. Equivariant attention interpretability: Can the authors provide qualitative analyses or visualizations of the learned attention patterns across Platonic frames? Such results could clarify whether the model learns distinct geometric filters or redundant orientations.

---

> ### Author Response · Authors · 2025-11-20
>
> We thank the reviewer for taking the time to evaluate our manuscript thoroughly and contributing to its improvement. We are glad to see that the reviewer appreciates the quality of our manuscript and empirical effectiveness of our approach across multiple domains. We address each of the reviewers' concerns separately below. We hope to continue the discussion if any concerns remain.
>
> ## **Weaknesses**
>
> #### **1. Nontrivial trade-off between group size and model expressivity**
> We have answered a similar question for reviewer VMfF for question 1. We would like to refer you to that answer.
>
> #### **2. Comparison with stronger baselines**
> In our work, we compare to current state-of-the-art EquiformerV2, which outperforms EquiformerV1 by the same authors. In the original EquiformerV1 work, the authors outperform against SE(3)-Transformers, which was adequately tuned for QM9. As Platonic Transformers outperform EquiformerV2, we believed it would be redundant to provide this result for SE(3)-Transformer. We present these collated results from across works in the table below:
>
> | Model | $\mu$ | $\alpha$ |
> | :--- | :--- | :--- |
> | SE(3)-Transformer | .051 | .142 |
> | EquiformerV1 | .011 | .046 |
> | EquiformerV2 | .010 | .050 |
> | **Platonic Transformer (Ours)** | **.010** | **.048** |
>
> We hope to compare against ERoPE (Frank et al., 2024) in the near future but are immediately unable to as the original implementation is in JAX whilst ours is in Torch. However, if we were to rely on intuition, ERoPE performs _symmetrization_ of its input point clouds (which we avoid) and also integrates over the sphere (using Lebedev Quadrature), which may introduce bottlenecks. A JAX-based implementation of Platonic Transformers is underway, and we thank the reviewer for suggesting this additional comparison.
>
> ## **Questions**
> #### **1. More details on the scaling of the Fourier implementation**
>
> To clarify, we report actual training throughput as the hidden dimension is scaled in Table 8.
> We could also benchmark throughput for a single Transformer block, but it will show essentially the same as Table 8.
> For comparison with standard Transformers we refer to the benchmarking in the main paper, Table 4, showing that Platonic Transformer blocks implemented in the spatial domain slightly lag behind the default PyTorch implementation (in our current implementation).
> When scaling up the hidden dimension, the Fourier implementation will be faster than standard Transformers eventually simply due to the lower amount of FLOPs.
> We leave it to future work to study the benefits of scaling Platonic Transformers in detail.
>
> #### **2. Equivariant attention interpretability**
>
> Thanks for the suggestion!
> We've added a visualization of the attention scores in Appendix M. While the attention mostly focusses on local areas, the different frames show distinct directional patterns, meaning the model doesn't just look at the immediate neighborhood.

---

> > ### Author Response · Authors · 2025-11-20
> >
> > We once again thank the reviewer for their diligent work and suggestions that have improved the manuscript. Should any further concerns or points of clarification remain, we would be pleased to continue this dialogue. If, however, all remaining questions have been addressed, we kindly request the reviewer to consider increasing the score assigned to the manuscript.

---

### Official Review · Reviewer_VMfF · 2025-11-03

**Soundness:** 3
**Presentation:** 3
**Contribution:** 2
**Rating:** 6
**Confidence:** 3

**Summary:**

The paper asks how to endow Transformers with strong geometric inductive bias—translation and (discrete) rotation/reflection equivariance—without the heavy machinery and overhead of conventional equivariant networks. The authors propose the Platonic Transformer, which “lifts” features to a group axis indexed by elements of a discrete subgroup $G \in O(3)$, then shares weights across these frames while keeping the rest of the Transformer (including RoPE) unchanged. Attention is computed per-frame and then aggregated, giving an equivariant architecture with the same computation graph as a standard Transformer.

**Strengths:**

Originality:
1.  “lift-and-share” design: introducing a group axis and letting ordinary RoPE-attention operate unchanged per reference frame, with equivariant weight sharing across frames.
2. Insightful dynamic-convolution perspective that clarifies RoPE’s inductive bias; pragmatic use of finite Platonic groups offers a tractable middle ground between full continuous equivariance and invariance.
Quality:
1.  Readable narrative from lifting to equivariant linears to attention; figures help (e.g., pipeline diagram and scaling plots).
2. Multi-domain evaluation (images, point clouds, molecules) supports general usefulness of the bias; explicit invariant vs. equivariant attention comparison.
Significance:
1. This method serves as a near drop-in replacement, preserving the original computation graph and standard modules. Furthermore, its linear-time complexity, achieved via a Fourier-domain implementation, ensures excellent scalability.

**Weaknesses:**

Limited Fundamental Novelty: While the combination of ideas is novel, the fundamental building blocks are well-known. The use of group convolutions to ensure equivariance in linear layers is a cornerstone of G-CNNs (Cohen & Welling, 2016). The concept of lifting features to a group and operating on them is also standard in this field. The attention mechanism itself is the standard RoPE-attention, but applied in parallel. The method can be viewed as an instance of a Message Passing Neural Network (MPNN) where messages are computed via attention over multiple "views" (the reference frames), and updates are performed by group-convolved MLPs. The contribution is thus more of a sophisticated and effective architectural design rather than the introduction of a fundamentally new principle of equivariance. This is not a fatal flaw, as the design is very clever, but it positions the work as an incremental (though important) step forward.

Analysis of Design Choices: The paper introduces several crucial design choices but could benefit from a more in-depth analysis of their impact.
1. The choice to fix key vectors ($k_j$ =1) in the linear attention variant is justified by observed training instability on molecular datasets. The hypothesis that this disentangles geometry and signal is interesting but remains a hypothesis. A more rigorous investigation into this instability and potential alternatives (e.g., regularization, different initializations for the key network) would strengthen this design claim.
The comparison between equivariant attention (Eq. 11) and invariant attention (Eq. 12) is only done theoretically. An empirical ablation study comparing these two would provide concrete evidence for the benefit of the more expressive, orientation-dependent attention patterns.

Missing related work on frame methods：
1. A new perspective on building efficient and expressive 3D equivariant graph neural networks，  Neurips 2024
2. AlphaNet: Scaling Up Local Frame-based Atomistic Foundation Model，  Npj CM, 2025

**Questions:**

\textbf{Heads vs. Group Size:} The framework maps group elements to attention heads. What is the interplay between the size of the group $|G|$ and the number of heads per group element `nhead`? For a fixed total number of heads ($|G| \times \text{nhead}$), have you explored the trade-off between using a larger, more expressive group (e.g., Octahedral, $|G|=24$) with `nhead=1` versus a smaller group (e.g., Tetrahedral, $|G|=12$) with `nhead=2`? This could shed light on whether it's more beneficial to have more geometric frames or more feature diversity per frame.

 \textbf{Stability of Learned Keys:} Could you please elaborate on the instability observed when using learned keys in the linear convolutional variant on QM9/OMol25? Did the training loss diverge, or did it just converge to a poor result? Is it possible that this instability is an optimization artifact that could be mitigated with, for example, a different learning rate, initialization, or a regularizer on the key-producing network, rather than a fundamental issue requiring the removal of learned keys?

 \textbf{Practical Computational Cost:} Table 4 shows impressive inference times compared to other geometric networks. However, the claim is that the cost is identical to a standard Transformer. Group convolutions (Eq. 8), when implemented in the spatial domain, may have different memory access patterns than a dense matrix multiplication. Could you provide a more direct wall-clock time comparison between a single layer of your spatial implementation and a standard \texttt{TransformerEncoderLayer} with an equivalent total feature dimension (i.e., \texttt{d\_model = |G| * d\_hidden}) and number of points, to precisely quantify any practical overhead?

\textbf{Invariant Attention Ablation:} The discussion in Section 4.2 contrasting equivariant and invariant attention scores is very clear. Given that implementing the invariant score via sum-pooling seems straightforward, was this variant tested? If so, how did it perform? If not, what is your intuition on how much performance would be lost by sacrificing orientation-dependent attention patterns?

---

> ### Author Response · Authors · 2025-11-20
>
> We thank the reviewer for their thorough assessment of our work, and are glad to see that the reviewer appreciates the quality of our readable narative, as well as the originality of the proposed "lift-and-share" design within a rope-base transformer drop-in replacement preserving the computation graph and standard modules with linear-time versions.
>
> The reviewer raises a number of valid weaknesses and has some questions, which we address below.
>
> ## **Weaknesses**
> #### **Limited Fundamental Novelty**
> We agree that our work builds upon established principles like group convolutions and feature lifting. Our primary contribution is not the invention of a new principle of equivariance, but rather a novel and practical architectural design that resolves the critical trade-off between equivariance and scalability. By integrating these principles while preserving the exact computation graph of a standard Transformer , our method achieves a powerful geometric bias without the typical computational overhead, making it a scalable, ”drop-in” solution.
>
> #### **Analysis of Design Choices**
> Since we got a similar question by reviewer Pkaq which we answered elaborately at question 2, we would like to refer you there.
>
> #### **Missing related work on frame methods**
>
> Thank you for bringing these relevant works to our attention. We have added these citations to our discussion of frame-based methods in the revised manuscript.
>
> ## **Questions**
>
> #### **1. Heads vs. Group Size**
> We agree with the reviewer that a more explicit ablation study on the effective number of heads, effective head-dimensionality, and the size of the group could provide valuable insights into whether it is more beneficial to have more geometric frames or greater feature diversity per frame.
>
> Our existing experiments on different datasets implicitly use a fixed head size and ablate over various group sizes. These results indicated that increasing the number of geometric frames is beneficial, although with diminishing returns. The diminishing return could be caused by the effective head-dim becoming too small. To test this we added an extra ablation to the QM9 experiment which will be done in three days. Below we already posted the table such that you can see how it would look like. We will add it to the appendix as wel.
>
> On the other hand, we would also like to stress that in our manuscript we attempted to compare our setup with standard ViT sizes. However, when utilizing the Platonic Transformer to solve specific tasks, we would not necessarily constrain the effective-head-dim to maximize the number of group elements. Such architectural decisions would always be based on the available resources and task-specific ablations.
>
> | Num group elem | Hidden-Dim | Hidden-Dim-G | Num-Heads | Effective-num-heads | Effective-head-dim | Performance alpha | Performance mu |
> |:--------------:|:----------:|:------------:|:---------:|:-------------------:|:------------------:|:-----------------:|:--------------:|
> | 1 (Trivial)    | 1152       | 1152         | 72        | 72                  | 16                 | 0.028             | 0.064          |
> | 12 (Tetrahedron) | 1152       | 96           | 72        | 6                   | 16                 | 0.012             | 0.049          |
> | 24 (Octahedron)  | 1152       | 48           | 72        | 3                   | 16                 | 0.010             | 0.048          |
> | 1 (Trivial)    | 1152       | 1152         | 48        | 48                  | 24                 |                   |                |
> | 12 (Tetrahedron) | 1152       | 96           | 48        | 4                   | 24                 |                   |                |
> | 24 (Octahedron)  | 1152       | 48           | 48        | 2                   | 24                 |                   |                |

---

> > ### Author Response · Authors · 2025-11-20
> >
> > #### **2. Stability of Learned Keys**
> >
> > To address your question regarding whether the instability was an optimization artifact or a fundamental issue, we conducted additional experiments on QM9 to reproduce the instabilities encountered when developing our method. We have added these results to the manuscript in a new appendix called "Experiments on Learned Key Projections". In this section we plot the learning curves and final results for various scenarios when using learned keys. In short we observe the following. **Divergence:** When using the full Attention mechanism, introducing learned keys leads to severe instability and training collapse around epoch 10, confirming that the issue is not merely poor convergence but active divergence. **Performance Gap:** In the linear Convolutional mode, where training remains stable, learned keys consistently underperform our fixed-key ($k=1$) baseline, yielding higher test error. **Regularization Check:** We performed a weight decay sweep to test if regularization could mitigate the instability. While high weight decay ($>10^{-4}$) stabilizes the training, the resulting performance remains significantly worse than the standard fixed-key configuration.
> >
> > We will improve the formatting of the figures (font sizes are small), but wanted to already share these insights. Does this additional appendix in your opinion provide sufficient evidence for the observed instability?
> >
> > #### **3. Practical Computational Cost**
> > We suspect the concern regarding "memory access patterns" stems from a misunderstanding of Equation 8. To clarify, this equation describes the pointwise linear layers (e.g., QKV projections, MLPs), not the attention mechanism itself.
> > We have revised **Section 3** to explicitly introduce the "flattened vector viewpoint." As detailed there, while Eq. 8 is theoretically a group convolution, we implement it as a **standard dense matrix-vector multiplication** ($y_i = W f_i$). The equivariance is enforced solely by imposing a weight-sharing pattern within $W$. Consequently, our implementation maintains the exact computation graph and regular memory access patterns of a standard Linear layer.
> > Regarding timings, **Table 4** already provides the requested comparison: it benchmarks our layer against a standard PyTorch `TransformerEncoderLayer` with equivalent feature dimensions, showing they are on the same order of magnitude. Furthermore, we discuss in **Appendix N** (Implementing Platonic Transformers in the Fourier Domain of Finite Groups) how this specific weight-sharing structure actually enables more efficient implementations at scale via the Fourier domain.
> > Does this clarification regarding the matrix-multiplication implementation resolve your concern about memory access patterns?
> >
> > #### **4. Invariant Attention Ablation**
> >
> > We agree that this is a relevant ablation. The best performing model on QM9-$\alpha$ in Table 3 is the octahedral Platonic Transformer with test MAE 0.010. Replacing the equivariant attention scores with invariant attention scores, we obtain a test MAE of 0.019.
> > This result is better than the trivial baseline (0.028) but worse than the result with equivariant attention scores.

---

> > > ### Author Response · Authors · 2025-11-20
> > >
> > > We again would like to thank the reviewer for the effort and their help in improving the manuscript. If any concerns remain, or if we could help to clarify anything we would like to continue the conversation. If all answers were answered clarifying the remaining concerns, we would like the reviewer to consider increasing the grade.

---

> > > > ### Author Response · Authors · 2025-12-03
> > > > **Finalised response on '1. Heads vs. Group Size'**
> > > >
> > > > To investigate the impact of head allocation on model performance, we conducted an additional ablation study varying the number of attention heads (Num-Heads) and the resulting effective head dimension (Eff-head-dim). As shown in Table 1, we compared configurations with 72 and 48 heads. Notably, we observed negligible performance differences between these settings across both trivial and geometric group configurations.
> > > >
> > > > We attribute this robustness to the high expressivity of the architecture. Given the model's substantial depth and width, it appears to be sufficiently parameterized to compensate for potential variations in expressivity caused by differing head allocations. While we hypothesize that the num_heads hyperparameter might exhibit a stronger impact in more constrained regimes (e.g., fewer layers or smaller hidden dimensions, significantly more data), validating this trend required a broader experimental sweep that was not feasible within the rebuttal timeline.
> > > >
> > > > | Num group elem | Hidden-Dim | Hidden-Dim-G | Num-Heads | Effective-num-heads | Effective-head-dim | Performance alpha | Performance mu |
> > > > |:--------------:|:----------:|:------------:|:---------:|:-------------------:|:------------------:|:-----------------:|:--------------:|
> > > > | 1 (Trivial)    | 1152       | 1152         | 72        | 72                  | 16                 | 0.028             | 0.064          |
> > > > | 12 (Tetrahedron) | 1152       | 96           | 72        | 6                   | 16                 | 0.012             | 0.049          |
> > > > | 24 (Octahedron)  | 1152       | 48           | 72        | 3                   | 16                 | 0.010             | 0.048          |
> > > > | 1 (Trivial)    | 1152       | 1152         | 48        | 48                  | 24                 |    0.027    |   0.064        |
> > > > | 12 (Tetrahedron) | 1152       | 96           | 48        | 4                   | 24                 |  0.012  |     0.048             |
> > > > | 24 (Octahedron)  | 1152       | 48           | 48        | 2                   | 24          |       0.011   |   0.047       |

---

### Author Response · Authors · 2025-12-03
**Final Response**

**General Response to All Reviewers and ACs**

We thank all involved for their time, thorough assessment, and constructive feedback. Especially considering the overhead caused by OpenReview this year.

We are encouraged by the consensus on the work's core strengths, specifically the **originality of the "lift-and-share" design**, the practicality of our method as a **scalable, efficient "drop-in" replacement** for standard Transformers, and the **clarity and readability** of the narrative. The review process has significantly strengthened our manuscript.

Overall, we believe this submission represents a strong contribution to the field. It demonstrates that geometric inductive biases can be integrated without the typical computational overhead. By designing our method as a direct **drop-in replacement**, we achieve **significant efficiency gains** and **improved or similar performance compared to standard Transformers**, offering a practical and scalable solution for geometric modeling.

Below is a summary of the specific changes, experiments, and clarifications added to the revised manuscript:

**1. New Experimental Results & Ablations:**
* **Stability of Learned Keys (New Appendix L):** To address Reviewer VMfF, we added a new appendix, "Experiments on Learned Key Projections." This includes learning curves and weight decay sweeps demonstrating that learned keys lead to training divergence or inferior performance compared to our fixed-key design, validating our architectural choice.
* **Heads vs. Group Size Trade-off (Final response reviewer VMfF):** We added an ablation study (shown in the response to Reviewer VMfF) analyzing the trade-off between the number of geometric frames and effective head dimensionality.
* **Invariant Attention Ablation (Response to VMfF):** We added a result comparing equivariant attention scores against invariant attention scores on QM9, quantifying the benefit of our approach.
* **QM9 Targets (Appendix J.4):** We have initiated runs for the remaining QM9 targets to provide a comprehensive evaluation as requested by Reviewer uQvE.

**2. Visualizations & Baselines:**
* **Attention Visualization (Appendix M):** Added visualizations of attention scores to illustrate how the model utilizes different geometric frames (addressed Reviewer Pkaq).
* **Baseline Comparisons (Appendix J.4):** Included a collated comparison table with SE(3)-Transformers and Equiformer V1/V2 to contextualize our performance against strong baselines (addressed Reviewer Pkaq).

**3. Clarifications & Textual Improvements:**
* **Finite vs. Continuous Equivariance:** We revised the text to explicitly state that our method provides equivariance to translations and **finite subgroups** of $O(3)$, rather than full continuous $SE(3)$ equivariance, and cited relevant literature on "relaxed equivariance" (addressed Reviewers rkTK & uQvE).
* **Implementation Details (Section 3):** We rewrote portions of Section 3 clarifying that Equation 8 is implemented as a standard dense matrix-vector multiplication. This addresses concerns regarding memory access patterns (addressed Reviewer VMfF).
* **Frame Averaging:** We expanded the discussion on why we avoid pre-concatenating frames (batch size/memory constraints) versus separate forward passes (addressed Reviewer uQvE).
* **Citations:** Added missing references regarding frame-based methods (addressed Reviewer VMfF).

We thank the reviewers again for their positive evaluation of the paper and for proposing the above improvements that further strengthened the impact of the paper.

---

### Meta-Review · Area_Chair_158X · 2026-01-05

**Summary:**

The submission introduces an equivariant transformer model that lifts features to discrete subgroups of O(3) and uses a separate attention head for each group element, allowing use of standard attention blocks and RoPE, thus incurring is less computational overhead than prior work.

Initial reviews were mixed. Reviewers valued the originality, simplicity, and efficiency of the method, as well as the good quality of exposition of the paper. Concerns were raised regarding missing strong baselines, unconvincing experiments, limited novelty, and lacking some ablations.

The rebuttal clarified some points and provided extra ablations, but unfortunately many necessary suggested experiments were not done: comparisons against ERoPE, comparisons against stronger baselines on vision tasks, fair comparisons on OMol25. The results added for the rest of QM9 targets were underwhelming.

The rebuttal mentions that many of the lacking experiments and comparisons are being worked on. Thus, resubmission with the new results seems more appropriate, and I recommend rejection at this time.

**Reviewer Concerns:**

The most serious concerns were about unconvincing experimental results. The rebuttal addressed the requested ablations and other minor concerns satisfactorily.

Pkaq requested comparison against ERoPE, which could not be provided on time.

rkTK requested comparison against other equivariant methods in vision tasks, which was not done. In fact, the submission results on CIFAR-10 seem worse than Weiler & Cesa (NeurIPS'19), who had already shown an equivariant model outperforming its non-equivariant counterpart many years ago.

uQvE requested the results for all QM9 targets, as well as a fair comparison against eSEN on OMol25, instead of cutting its training short without changing its hyperparameters. The extra results on QM9 were provided, and it seems that all underperform the several of the baselines. The fair comparison against eSEN was not provided.

**Reviewer Scores:**

Given the remaining concerns listed above, I believe Pkaq, rkTK, uQvE would maintain or reduce their scores. VMfF was already favorable to the submission and their concerns were addressed, so they would likely maintain or increase the score.

---

### Decision · Program_Chairs · 2026-01-26

Reject